# Online Matching in Sparse Random Graphs: Non-Asymptotic Performances of Greedy Algorithm

**Nathan Noiry**
Télécom Paris, Palaiseau, France
`noirynathan@gmail.com`

**Flore Sentenac**
CREST, ENSAE Paris, Palaiseau, France
`flore.sentenac@ensae.fr`

**Vianney Perchet**
CREST, ENSAE Paris, Palaiseau, France
CRITEO AI Lab, Paris, France
`vianney.perchet@normalesup.org`

## Abstract

Motivated by sequential budgeted allocation problems, we investigate online matching problems where connections between vertices are not i.i.d., but they have fixed degree distributions – the so-called configuration model. We estimate the competitive ratio of the simplest algorithm, GREEDY, by approximating some relevant stochastic discrete processes by their continuous counterparts, which are solutions of an explicit system of partial differential equations. This technique gives precise bounds on the estimation errors, with arbitrarily high probability as the problem size increases. In particular, it allows the formal comparison between different configuration models. We also prove that, quite surprisingly, GREEDYcan have better performance guarantees than RANKING, another celebrated algorithm for online matching that usually outperforms the former.

## 1 Introduction

Finding matchings in bipartite graphs $(\mathcal{U} \cup \mathcal{V}, \mathcal{E})$, where $\mathcal{E} \subset \mathcal{U} \times \mathcal{V}$ is a set of edges, is a long-standing problem with different motivations and approaches [Godsil, 1981, Zdeborová and Mézard, 2006, Lovász and Plummer, 2009, Bordenave et al., 2013]. If $\mathcal{U}$ is seen as a set of resources and $\mathcal{V}$ as demands, the objective is to allocate as many resources to demands (an allocation - or a matching - between $u$ and $v$ is admissible if $(u, v) \in \mathcal{E}$) with the constraint that a resource is allocated to only one demand and vice-versa.

Motivated particularly by practical applications to Internet advertising, the *online* variant of this problem is receiving increasing attention (we refer to the excellent survey [Mehta, 2012] for more applications, specific settings, results and techniques). In this case, the set of vertices $\mathcal{U}$ is present at the beginning and the graph unveils sequentially: vertices $v \in \mathcal{V}$ are observed sequentially, one after the other, along with the edges they belong to. An online algorithm must decide, right after observing $v_k$ and its associated set of edges $\mathcal{E}_k := \{(u, v_k) \in \mathcal{E}\}$ to match it to some other vertex $u \in \mathcal{U}$, at the conditions that $(u, v_k) \in \mathcal{E}_k$ and $u \in \mathcal{U}$ has not been matched yet. The performance of an online algorithm is evaluated by its *competitive ratio*, which is the ratio between the size of the matching it has created and the highest possible matching in hindsight [Feldman et al., 2009].

This theoretical setting is particularly well suited for online advertising: $\mathcal{U}$ is the set of campaigns/ads that an advertiser can run and users $v_1, v_2, \ldots, v_T$ arrive sequentially [Mehta, 2012, Manshadi et al., 2012]. Some of them are eligible for a large subset of campaigns, others are not (usually based

35th Conference on Neural Information Processing Systems (NeurIPS 2021).

on their attributes/features, such as the geographic localization, the browsing history, or any other relevant information). The objective of an advertiser (in this over-simplified model) is to maximize the number of displayed ads. In practice, campaigns/ads are not displayed only once but have a maximal budget of impressions (say, a specific ad can be displayed only 10.000 times each day). A possible trick consists of duplicating the vertices of $\mathcal{U}$ as many times as the budget. However, this results in strong and undesirable correlations between vertices. It is, therefore, more appropriate to consider a bipartite graph with *capacities* and admissible matchings as subsets of edges such that each vertex belongs to several different edges, but not more than their associated capacities $\omega \in \mathbb{N}$ (a vertex $v \in \mathcal{V}$ is matched once while $u \in \mathcal{U}$ can be matched $\omega_u$ times).

This online matching problem with capacities has been quite extensively studied. It is known that GREEDY, which matches all incoming vertices to any available neighbor has a competitive ratio of $1/2$ in the worst case, albeit it achieves $1 - 1/e$ as soon as the incoming vertices arrive in Random Order [Goel and Mehta, 2008b]. The worst-case optimal algorithm is the celebrated RANKING, which achieves $1 - 1/e$ on any instance [Karp et al., 1990, Devanur et al., 2013, Birnbaum and Mathieu, 2008], and also has better guarantees in the Random Order setting [Mahdian and Yan, 2011].

Beyond the adversarial setting, the following stochastic setting has been considered: there exist a finite set of $L$ "base" vertices $v^{(1)}, \ldots, v^{(L)}$ associated to base edge-sets $\mathcal{E}^{(1)}, \ldots, \mathcal{E}^{(L)}$. When a vertex $v_k$ arrives, its type $\theta_k \in \{1, \ldots, L\}$ is drawn iid from some distribution (either known beforehand or not) and then its edge set is set as $\mathcal{E}_k = \mathcal{E}^{(\theta_k)}$. In the context where the distribution is known, algorithms with much better competitive ratios than GREEDY or RANKING were designed [Manshadi et al., 2012, Jaillet and Lu, 2014, Brubach et al., 2019], specifically with a competitive ratio of $1 - 2/e^2$ when the expected number of arrival of each type are integral and $0.706$ without this assumption. Notably, those competitive ratios still hold with Poisson arrival rates rather than a fixed number of arrivals.

On a side note, a vast line of work considers online matching in weighted graphs [Devanur et al., 2012, Goel and Mehta, 2008a, Mehta, 2012], which is outside the scope of this paper. However, it is still worth noting that the unweighted graph is a weighted graph with all weights equal.

This model of the stochastic setting is quite interesting but rather strong: it lacks flexibility and cannot be used to represent some challenging instances ( for example when the degrees of each vertex $\mathcal{U}$ increase linearly with the number of vertices in $\mathcal{V}$, or when the set $\mathcal{U}$ of campaigns must be fixed so that the model is well specified, etc...). Another tentative is to consider Erdős-Rényi graphs assuming that each possible edge is present in $\mathcal{U} \times \mathcal{V}$ with some fixed probability and independently of the other edges (see [Mastin and Jaillet, 2013]). The most interesting and challenging setting corresponds to the so-called *sparse* regime where each vertex of $\mathcal{U}$ has an expected degree independent of the size $n$ of $\mathcal{V}$, which amounts to take a probability of connection equal to $c/n$. Interestingly enough, even the analysis of the simplest GREEDY algorithm is quite challenging and already insightful in those models [Borodin et al., 2018, Arnosti, 2019, Dyer et al., 1993, Mastin and Jaillet, 2013]. Unfortunately, although this Erdős-Rényi model is compatible with growing sets $\mathcal{U}$ and $\mathcal{V}$, it also turns out to be quite restrictive. The main reason is that the approximate Poisson degree distribution of the vertices has light-tail and does not allow for the appearance of the so-called *scale-free property* satisfied by many real-world networks [Barabási et al., 2000, Van Der Hofstad, 2016].

We, therefore, consider a more appropriate random graphs generation process called *configuration model*, introduced by [Bender and Canfield, 1978] and [Bollobás, 1980]. The optimal matching of this model has been computed in [Bordenave et al., 2013]. The configuration model is particularly well suited to handle different situations such as the following one. Assume that campaigns can either be "intensive" (with many eligible users) or "selective/light" (few eligible users), with an empirical proportion of, say, 20%/80%. Then whether an advertiser handles 100 campaigns at the same time or 10.000, it will always have roughly this proportion of intensive vs. light campaigns. Similarly, some users are more valuable than others, and are thus eligible for more campaigns than the others; the proportion of each type being independent of the total population size. The configuration model accommodates these observations by basically drawing iid degrees for vertices $\mathcal{U}$ and $\mathcal{V}$ (accordingly to some different unknown distributions for $\mathcal{U}$ and $\mathcal{V}$) and then by finding a graph such that those degrees distribution are satisfied (up to negligible errors); as a consequence, the graphs generated are *sparse*, in the sense that the number of edges grows linearly with the number of vertices.

Additionally, the configuration model is a well-suited random graph model which mimics a number of properties of real-world complex networks, while being analytically tractable. For instance, choosing

power-law distributions for the degrees allows to obtain the so-called scale-free property (often observed in practice, as highlighted for the web by Faloutsos et al. [1999]). The configuration model also displays the so called "small-world phenomenon" (observed for instance in the graph of Facebook by Backstrom et al. [2012]) as its diameter is of logarithmic order.

**Main contribution**

We investigate the performances (in terms of expected competitive ratio) of the GREEDY matching algorithm in configuration models and we provide explicit quantitative results using stochastic approximation techniques [Wormald, 1995]; we prove that the increasing size of the random matching created is arbitrarily close to the solution of some explicit ODE. Solving the latter then gives in turn the solution to the original problem.

The remaining of the paper is organized as follows. Section 2 describes precisely the problem and Theorem 1 is our first main result: it describes the performances of GREEDY in the capacity-less problem. The proof of Theorem 1 is delayed to Appendix D, but the main ideas and intuitions are provided in Section 3. The online matching with capacities problem is treated in Appendix A.

## 2 Online Matching Problems; Models and main result

Consider a bipartite graph with capacities $G = (\mathcal{U}, \mathcal{V}, \mathcal{E}, \omega)$ where $\mathcal{U} = \{1, \ldots, N\}$ and $\mathcal{V} = \{1, \ldots, T\}$ are two finite set of vertices, $\mathcal{E} \subset \{(u, v), u \in \mathcal{U}, v \in \mathcal{V}\}$ is the set of edges and $\omega : \mathcal{U} \to \mathbb{N}_*$ is a capacity function. A matching $M$ on $G$ is a subset of edges $e \in \mathcal{E}$ such that any vertex $v \in \mathcal{V}$ is the endpoint of at most one edge $e \in M$ and any vertex $u \in \mathcal{U}$ is the endpoint of at most $\omega_u$ edges in $M$. We will denote by $\mathcal{M}$ the set of matchings on $G$; the optimal matching $M^* \in \mathcal{M}$ is the one (or any one) with the highest cardinality, denoted by $|M^*|$.

The batched matching problem consists in finding any optimal matching $M^*$ given a graph with capacities $G$; the online variant might be a bit more challenging, as the matching is constructed sequentially. Formally, the set of vertices $\mathcal{U}$ and their capacities $\omega$ are known from the start, and vertices $v \in \mathcal{V}$ arrive sequentially (with the edges they belong to) and $M_0 = \emptyset$. At stage $t \in \mathbb{N}$ – assuming a matching $M_{t-1}$ has been constructed –, a decision maker observes a new vertex[1] $v_t$ and its associated set of edges $\{(u, v_t); u \in \mathcal{E}\}$. If possible, one of these edges $(u_t, v_t)$ is added to $M_{t-1}$, with the constraint that $M_t = M_{t-1} \cup \{(u_t, v_t)\}$ is still a matching. The objective is to maximize the size of the constructed matching $M_T$. The classical way to evaluate the performances of an algorithm is the *competitive ratio*, defined as $|M_T|/|M^*| \in [0, 1]$ (the higher the better).

### 2.1 Structured online matching via Configuration Model

As mentioned before, the online matching problem can be quite difficult without additional structure. We will therefore assume that the vertex degrees in $\mathcal{U}$ and $\mathcal{V}$ have (at least asymptotically in $N$ and $T$) some given subGaussian[2] distributions $\pi_{\mathcal{U}}$ and $\pi_{\mathcal{V}}$, of respective expectation $\mu_{\mathcal{U}}$ and $\mu_{\mathcal{V}}$ and respective proxy-variance $\sigma_{\mathcal{U}}^2$ and $\sigma_{\mathcal{V}}^2$. Those numbers are related in the sense that we assume[3] that $T = \frac{\mu_{\mathcal{U}}}{\mu_{\mathcal{V}}} N \in \mathbb{N}$. Given those degree distributions, the graphs we consider are random draws from a bipartite configuration model described below; for the sake of clarity, we first consider the capacity-less case (when $\omega_u = 1$ for all $u \in \mathcal{U}$).

Given $\pi_{\mathcal{U}}$ and $\pi_{\mathcal{V}}$ and $N, T \geq 1$, let $d_1^{\mathcal{U}}, \ldots, d_N^{\mathcal{U}} \in \mathbb{N} \overset{\text{i.i.d.}}{\sim} \pi_{\mathcal{U}}$ and $d_1^{\mathcal{V}}, \ldots, d_T^{\mathcal{V}} \in \mathbb{N} \overset{\text{i.i.d.}}{\sim} \pi_{\mathcal{V}}$ be independent random variables; intuitively, those numbers are respectively the number of half-edges attached to vertex in $\mathcal{U}$ and $\mathcal{V}$. Consider also two extra random variables

$$d_{T+1}^{\mathcal{V}} = \max \Big\{ \sum_{i=1}^{N} d_i^{\mathcal{U}} - \sum_{j=1}^{T} d_j^{\mathcal{V}}, 0 \Big\} \quad \text{and} \quad d_{N+1}^{\mathcal{U}} = \max \Big\{ \sum_{j=1}^{T} d_j^{\mathcal{V}} - \sum_{i=1}^{N} d_i^{\mathcal{U}}, 0 \Big\}$$

---

[1] Although the order of arrival is irrelevant to the models we studied, it could have an impact on other models.

[2] $X$ is subGaussian with proxy-variance $\sigma^2$ if for any $s \in \mathbb{R}, \mathbb{E}[\exp(sX)] \leq \exp\left(\frac{\sigma^2 s^2}{2}\right)$. Actually, we only need that $\pi_{\mathcal{U}}$ and $\pi_{\mathcal{V}}$ have some finite moment of order $\gamma > 2$.

[3] In the general case, consider $T = \lfloor N\mu_{\mathcal{U}}/\mu_{\mathcal{V}} \rfloor$. The proof is identical, up to a negligible $1/N$ error term

so that equality between total degrees holds, i.e., $\sum_{i=1}^{N+1} d_i^{\mathcal{U}} = \sum_{j=1}^{T+1} d_j^{\mathcal{V}}$. Finally, a random (capacity-less) bipartite graph denoted by $\mathbf{CM}(\mathbf{d}^U, \mathbf{d}^V)$ is constructed with a uniform pairing of half-edges of $\mathcal{U} \cup \{N+1\}$ with half-edges of $\mathcal{V} \cup \{T+1\}$ and removing vertices $T+1$ and $N+1$ and their associated edges. These two artificially added vertices are just here to define a pairing between half-edges. Notice that, by the law of large numbers and since $T = (\mu_{\mathcal{U}}/\mu_{\mathcal{V}})N$, $d_{T+1}^{\mathcal{V}} = o(N)$ and $d_{N+1}^{\mathcal{U}} = o(N)$ almost surely[4].

The bipartite configuration model $\mathbf{CM}(\mathbf{d}^{\mathcal{U}}, \mathbf{d}^{\mathcal{V}})$ is then the random graph obtained by a uniform matching between the half-edges of $\mathcal{U}$ and the half-edges of $\mathcal{V}$, where the random sequences $\mathbf{d}^{\mathcal{U}} = (d_i^{\mathcal{U}})_i$ and $\mathbf{d}^{\mathcal{V}} = (d_j^{\mathcal{V}})_j$ are defined as above.

## 2.2 Competitive ratio of GREEDY algorithm. Main result

The first question to investigate in this structured setting is the computation of the (expected) competitive ratio of the simple algorithm GREEDY. It constructs a matching by sequentially adding any admissible edge uniformly at random. Describing it and stating our results require the following additional notations: for any $e = (u, v) \in E$, $u(e) = u$ (resp. $v(e) = v$) is the extremity of $e$ in $\mathcal{U}$ (resp. $\mathcal{V}$); the generating series of $\pi_{\mathcal{U}}$ and $\pi_{\mathcal{V}}$ are denoted by $\phi_{\mathcal{U}}$ and $\phi_{\mathcal{V}}$ and are defined as

$$\phi_{\mathcal{U}}(s) := \sum_{k \geq 0} \pi_{\mathcal{U}}(k) s^k \qquad \text{and} \qquad \phi_{\mathcal{V}}(s) := \sum_{k \geq 0} \pi_{\mathcal{V}}(k) s^k.$$

Our first main theorem, stated below, identifies the asymptotic size of the matching generated by GREEDY on the bipartite configuration model we have just defined. As the batched problem (i.e., computing the size of the optimal matching $M^*$) is well understood [Bordenave et al., 2013], this quantity is sufficient to derive competitive ratios. Again, for the sake of presentation, we first assume that all capacities are fixed, equal to one; the general case is presented in Appendix A.

**Theorem 1.** *(Performances of* GREEDY *in the capacity-less case)*

*Given $N \geq 1$ and $T = \frac{\mu_{\mathcal{U}}}{\mu_{\mathcal{V}}} N$, let $M_T$ be the matching built by* GREEDY *on $\mathbf{CM}(\mathbf{d}^U, \mathbf{d}^V)$ then the following convergence in probability holds:*

$$\frac{|M_T|}{N} \xrightarrow[N \to +\infty]{\mathbf{P}} 1 - \phi_{\mathcal{U}}(1 - G(1)).$$

*where $G$ is the unique solution of the following ordinary differential equation:*

$$G'(s) = \frac{1 - \phi_{\mathcal{V}}\left(1 - \frac{1}{\mu_{\mathcal{U}}}\phi_{\mathcal{U}}'\left(1 - G(s)\right)\right)}{\frac{\mu_{\mathcal{V}}}{\mu_{\mathcal{U}}}\phi_{\mathcal{U}}'(1 - G(s))}; \quad G(0) = 0. \tag{1}$$

*Moreover, for any $s \in [0,1]$, if $M_T(s)$ is the matching obtained by* GREEDY *after seeing a proportion $s$ of vertices of $\mathcal{V}$, then*

$$\frac{|M_T(s)|}{N} \xrightarrow[N \to +\infty]{\mathbf{P}} 1 - \phi_{\mathcal{U}}(1 - G(s)). \tag{2}$$

*Convergence rates are explicit; with probability exponentially large, at least $1 - \zeta N \exp(-\xi N^{c/2})$,*

$$\sup_{s \in [0,1]} \left| \frac{|M_T(s)|}{N} - \left(1 - \phi_{\mathcal{U}}(1 - G(s))\right) \right| \leq \kappa N^{-c},$$

*where $\zeta, \xi, \kappa$ depend only on the (first two) moments of both $\pi_{\mathcal{V}}$ and $\pi_{\mathcal{U}}$, and $c$ is some universal constant (set arbitrarily as $1/20$ in the proof).*

Theorem 1 generalizes to the case with capacities, see Sections A.1 and A.2. The details of the proof of Theorem 1 are postponed to Appendix D, but the main ideas are given in Section 3.

## 2.3 Examples, Instantiations and Corollaries

We provide in this section some interesting examples and corollaries that illustrate the powerfulness of Theorem 1, and how it can be used to compare different situations.

---

[4]And even $\mathcal{O}(\sqrt{N})$ with probability exponentially large in $N$ as both distributions are sub-Gaussian. So the effects of those additional vertices can be neglected.

### 2.3.1 $d$-regular graphs

The first typical example of random graphs are " $d$-regular ", for some $d \in \mathbb{N}$, i.e., graphs such that each vertex has an exact degree of $d$ (to avoid trivial examples, we obviously assume $d \geq 2$).

It is non-trivial to sample a $d$-regular graph at random, yet it is easy to generate random graphs $G_N$ with the configuration model described above, with the specific choices of $\pi_{\mathcal{U}} = \pi_{\mathcal{V}} = \delta_d$, the Dirac mass at $d$. The downside is that $G_N$ is not exactly a $d$-regular bipartite random graph (as some vertices might be connected more than once, i.e., there might exist parallel edges). However, conditioned to be *simple*, i.e, without multiple edges and loops, it has the law of a uniform $d$-regular bipartite random graph. Moreover, the probability of being simple is bounded away from 0 [Van Der Hofstad, 2016]; as a consequence, any property holding with probability tending to 1 for $G_N$, holds with probability tending to 1 for uniform $d$-regular bipartite random graphs. Finally, we also mention that Hall's Theorem [Frieze and Karoński, 2016] implies that $G_N$ admits a perfect matching, so that $|M^*| = N$.

Instantiating Equation (1) to $d$-regular graphs yields that the competitive ratio of GREEDY converges, with probability 1, to $1 - (1 - G(1))^d$ where $G$ is the solution of the following ODE

$$\frac{(1 - G(s))^{d-1}}{1 - (1 - (1 - G(s))^{d-1})^d} G'(s) = \frac{1}{d}. \tag{3}$$

As expected, had we taken $d = 1$, then $G(s) = s$ hence the competitive ratio of GREEDY is 1 (but again, $d = 1$-regular graphs are trivial). More interestingly, if $d = 2$, the ODE has a closed form solution: $G(s) = \exp(\frac{s}{2}) - 1$, so that the competitive ratio of GREEDY converges to $4\sqrt{e} - (e + 3) \simeq 0.877 \gg 1 - \frac{1}{e} \simeq 0.632$, where the latter is a standard bound of the competitive ratio of GREEDY (for general, non-regular graphs) [Mehta, 2012].

**Solving Equation** (3)  In the general case $d \geq 3$, even if Equation (3) does not have a closed form solution, it is still possible to provide some insights. Notice first that the polynomial $P(X) = 1 - (1 - (1 - X)^{d-1})^d$ admits $n := d(d - 1)$ roots, among which there is 1 with multiplicity $d - 1$. If $X$ is another root, then

$$\left(1 - (1 - X)^{d-1}\right)^d = 1 \iff 1 - (1 - X)^{d-1} = e^{\frac{ik\pi}{d}}, \ k = 1, \ldots, d - 1.$$

Therefore,

$$(1 - X)^{d-1} = 1 - e^{\frac{ik\pi}{d}},$$

which admits $d - 1$ distinct solutions for each $k = 1, \ldots, d - 1$. The resulting $n := (d - 1)^2$ distinct complex, denoted $x_1, \ldots, x_n$, are the roots of $P(X)/(1 - X)^{d-1}$, so the ODE reduces to:

$$\frac{y'(t)}{\prod_{1 \leq i \leq n} y(t) - x_i} = \frac{1}{d}. \tag{4}$$

Since the following trivially holds:

$$\frac{1}{\prod_{1 \leq i \leq n}(X - x_i)} = \sum_{1 \leq i \leq n} \frac{1}{\prod_{j \neq i}(x_i - x_j)} \frac{1}{X - x_i} =: \sum_{1 \leq i \leq n} \frac{a_i}{X - x_i}.$$

it is possible to integrate Equation (4) in $\sum_{1 \leq i \leq n} a_i \log(y(t) - x_i) = \frac{s}{d} + c$ to finally get

$$\prod_{1 \leq i \leq n}(y(t) - x_i)^{a_i} = C \exp(\frac{s}{d}),$$

and since $y(0) = 0$, it must hold that $C = \prod_{1 \leq i \leq n}(-x_i)^{a_i}$. As a consequence, $y(1)$ solves:

$$\prod_{1 \leq i \leq n}(y(1) - x_i)^{a_i} = e^{1/d} \prod_{1 \leq i \leq n}(-x_i)^{a_i}.$$

Unfortunately, even for $d = 3$, the solution somehow simplifies but has no closed form; on the other hand, numerical computations indicate that the competitive ratio of GREEDY converges to 0.89 when $d = 3$ and $N$ tends to infinity. We provide in Figure 3 the numerical solutions of the ODE for $d$-regular graphs (actually, we draw the functions $1 - \phi_U(1 - G(s))$ that are more relevant) for various values of $d$; the end-point obtained at $s = 1$ indicates the relative performance of GREEDY. As expected, those functions are point-wise increasing with $d$ (as the problem becomes simpler and simpler for GREEDY when $d \geq 2$).

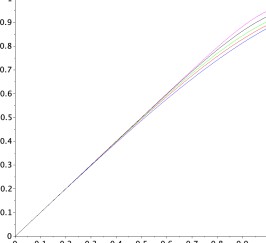 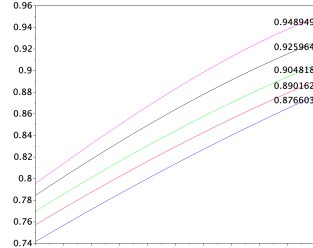

Figure 1: Numerical computations (on Scilab, results are almost instantaneous) of GREEDY performances for $d = 2$ (blue), $d = 3$ (red), $d = 4$ (green), $d = 6$ (black) and $d = 10$ (magenta). On the left, global solution, on the right, zoom-in on the end points with final values.

### 2.3.2 The Erdős-Rényi case.

In a Erdős-Rényi graph, there is an edge between two vertices $u \in \mathcal{U}$ and $v \in \mathcal{V}$ with some probability $p = \frac{c}{N}$, independently from each others. As $N$ goes to infinity, the number of edges to a vertex follows (approximately) a Poisson law of parameter $c > 1$.

As a consequence, we consider the configuration model where $\pi_{\mathcal{U}}$ and $\pi_{\mathcal{V}}$ are Poisson laws of parameter $c$, which yields $\mu = c$, $\phi_{\mathcal{U}}(s) = \mathrm{e}^{c(s-1)}$. In this case, Equation (1) becomes:

$$\frac{cG'(s)\,\mathrm{e}^{-cG(s)}}{1 - \mathrm{e}^{-c\,\mathrm{e}^{-cG(s)}}} = 1.$$

The solutions are given by:

$$G(s) = \frac{1}{c}\log\left(\frac{c}{\log(\mathrm{e}^{k-cs}+1)}\right),$$

yielding

$$\phi_X(1 - G(s)) = \frac{1}{c}\log\left(\mathrm{e}^{k-cs}+1\right).$$

The initial condition $\phi_{\mathcal{U}}(1 - G(0)) = \phi_{\mathcal{U}}(1) = 1$ gives $e^k = e^c - 1$, from which we deduce that the number of matches of GREEDY is asymptotically proportional to

$$1 - \phi_{\mathcal{U}}(1 - G(1)) = 1 - \frac{\log\left(2 - \mathrm{e}^{-c}\right)}{c},$$

which recovers, as a sanity check, some existing results [Mastin and Jaillet, 2013].

### 2.3.3 The comparison of different configuration models

Using Gronwall's Lemma, it is possible to show Theorem 1 can be used to compare different configuration models, as in the following Corollary.

**Corollary 1.** *Consider two configuration models* $\mathbf{CM}_1(\mathbf{d}_1^U, \mathbf{d}_1^V)$ *and* $\mathbf{CM}_2(\mathbf{d}_2^U, \mathbf{d}_2^V)$, *s.t.* $\mathbf{d}_1^U$ *and* $\mathbf{d}_1^U$ *are both drawn i.i.d. from* $\pi_U$, $\mathbf{d}_1^V$ *is drawn i.i.d. from* $\pi_V^1$ *and* $\mathbf{d}_2^V$ *is drawn i.i.d. from* $\pi_V^2$, *with* $\sum_x x\pi_V^1(x) = \sum_x x\pi_V^2(x)$. *If* $\phi_V^1(s) \geq \phi_V^2(s)$ *for any* $s \in (0,1)$, *then by denoting respectively* $\gamma_1$ *and* $\gamma_2$ *the asymptotic proportion of vertices matched by* GREEDY *in* $\mathbf{CM}_1(\mathbf{d}_1^U, \mathbf{d}_1^V)$ *and* $\mathbf{CM}_2(\mathbf{d}_2^U, \mathbf{d}_2^V)$, *it holds that necessarily* $\gamma_2 \geq \gamma_1$.

For instance, let us assume that the degree distribution on the offline side is fixed. Then the matching size obtained by GREEDY is asymptotically larger if vertices on the online side all have exactly the same degree $d$ rather than if those degrees are drawn from a Poisson distribution with expectation $d$.

A similar result (with a different criterion) holds with fixed degree distribution on the online side and differing one on the offline side.

### 2.4 GREEDY can outperform RANKING !

We recall that the RANKING algorithm, which is the worse case optimal, chooses at random a ranking over $\mathcal{U}$ and uses it to break ties (i.e., if two vertices $u$ and $u'$ can be matched to $v_k$, then it is the one with the smallest rank that is matched by RANKING). Quite surprisingly, we get that in the configuration model RANKING can have a worse competitive ratio than GREEDY, which advocates again for its thorough study.

**Proposition 1.** *Let $\gamma_R$ and $\gamma_G$ be the assymptotic performances of* RANKING *and* GREEDY *on the 2-regular graph. The following holds:*
$$\gamma_G > \gamma_R.$$
*In other words,* GREEDY *outperforms* RANKING *in the 2-regular graph.*

We conjecture that the above result actually holds for any $d \geq 2$, and more generally for a wide class of distributions $\pi_{\mathcal{U}}$ and $\pi_{\mathcal{V}}$ (finding a general criterion would be very interesting). The proof of Proposition 1 is provided in Appendix G. The main idea is that in the 2-regular graph, RANKING is biased towards selecting as matches vertices with two remaining half-edges rather than just one. Indeed, vertices with only one remaining half-edge were not selected previously and thus have a higher rank. The vertices with only one remaining half-edge will not get matched in the subsequent iterations, so not picking them as matches is suboptimal. On the other hand, GREEDY picks any match uniformly at random and does not exhibit such bias.

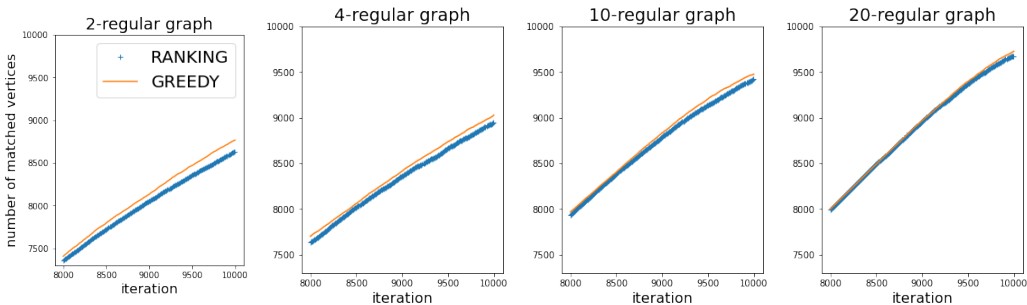

Figure 2: Experimental performances of GREEDY vs. RANKING on $d$-regular graphs

## 3 Ideas of proof of Theorem 1

The main idea behind the proof of Theorem 1 (postponed to Section D) is to show that the random deterministic evolution of the matching size generated by GREEDY is closely related to the solution of some ODE (this is sometimes called "the differential equation method" [Wormald, 1995] or "stochastic approximations" [Robbins and Monro, 1951]). Computing the solution of the ODE is easier - if not explicitly, at least numerically in intricate cases - than estimating the performances of GREEDY by Monte-Carlo simulations and it provides qualitative, as well as quantitative, properties.

Tracking the matching size is non-trivial because the vertices (in $\mathcal{U}$ and $\mathcal{V}$) have different degrees, hence some of them are more likely to be matched than others. However, in the configuration model, each vertex has the same distribution of degrees before the sequences $\mathbf{d}^{\mathcal{U}}$ and $\mathbf{d}^{\mathcal{V}}$ are fixed. As a consequence, the proof relies on the three following techniques

1. The graph is built sequentially, along with the matching and not beforehand (fixing the "randomness" at the beginning would be very difficult to handle in the analysis). Thankfully, this does not change the law of the graph generated (this is obviously crucial).

2. We are not only going to track the size of the matching built as we need to handle different probabilities of matching (and pairing the graph) for each vertex. As a consequence, we are going to track the numbers of non-matched vertices which have still $i$ half-edges to be paired and the number of already matched vertices that have $j$ half-edges remaining. This will give one different ODE per value of $i$ of $j$.

   Since $\pi_{\mathcal{U}}$ and $\pi_{\mathcal{V}}$ are sub-Gaussian, we will prove that with arbitrarily high probability - exponential in $N$ -, there are only a polynomial number of such equations

3. All those differential equations are then "aggregated" to build the final ODE satisfied by the matching size. Interestingly, this aggregated ODE has a simple form, while the full system is on the other hand quite intricate.

In the following sub-sections, we separate the proofs in the different building blocks to provide intuitions; the proof of technical lemmas are deferred to the appendix.

## 3.1 Building the graph together with the matching

The first step in the analysis is to notice that the bipartite configuration model can be constructed by sequentially pairing the half-edges coming from $\mathcal{V}$. The matching generated by GREEDY is then constructed simultaneously with the graph. More precisely, given two sequences[5] of non-negative integers $\mathbf{d}^{\mathcal{U}} = (d_1^{\mathcal{U}}, \ldots, d_N^{\mathcal{U}})$ and $\mathbf{d}^{\mathcal{V}} \cup \{d_{T+1}^{\mathcal{V}}\} = (d_1^{\mathcal{V}}, \ldots, d_T^{\mathcal{V}}, d_{T+1}^{\mathcal{V}})$, we introduce in the following a generating algorithm that simultaneously build the associated bipartite configuration model $\mathbf{CM}(\mathbf{d}^{\mathcal{U}}, \mathbf{d}^{\mathcal{V}})$ together with GREEDY. Recall that the bipartite configuration model is obtained through a uniform matching between the half-edges of $U$ and the half-edges of $V$. To avoid confusion, we will call a *marked matching* a pairing of two half-edges that corresponds to an edge that will belong to the constructed matching M. This construction pseudo-code is detailed in Algorithm 1.

---

**Algorithm 1:** GREEDY MATCHING CONFIGURATION MODEL WITHOUT CAPACITIES

---
**Input:** $\mathbf{d}^{\mathcal{U}} = (d_1^{\mathcal{U}}, \ldots, d_N^{\mathcal{U}})$ and $\mathbf{d}^{\mathcal{V}} = (d_1^{\mathcal{V}}, \ldots, d_T^{\mathcal{V}})$
**Initialization.** $M_0 \leftarrow \emptyset$, $\mathcal{E}_0 \leftarrow \emptyset$ and $H_0^{\mathcal{U}} \leftarrow \{$ half-edges of $\mathcal{U}\}$
**for** $t = 1, \ldots, T$ **do**
    Order uniformly at random the edges emanating from $v_t$: $e_1^t, \ldots, e_{k_t}^t$
    **for** $i = 1, \ldots, k_t$ **do**
        Choose uniformly an half-edge $e_i^{\mathcal{U}}$ in $H^{\mathcal{U}}$
        $\mathcal{E} \leftarrow \mathcal{E} \cup \{u(e_i^{\mathcal{U}}), v_t\}$              // Create an edge between $e_i^t$ and $e_i^{\mathcal{U}}$
        $H^{\mathcal{U}} \leftarrow H^{\mathcal{U}} \setminus \{e_i^{\mathcal{U}}\}$                  // Remove the half-edge
        **if** $v_t$ and $u(e_i^{\mathcal{U}})$ *unmatched* **then**
            $M_t \leftarrow M_{t-1} \cup \{u(e_i^{\mathcal{U}}), v_t\}$             // $v_t$ is matched
        **end**
    **end**
**end**
$\mathbf{CM}(\mathbf{d}^{\mathcal{U}}, \mathbf{d}^{\mathcal{V}}) \leftarrow (\mathcal{U}, \mathcal{V}, \mathcal{E})$.
**Output:** Bipartite configuration model $\mathbf{CM}(\mathbf{d}^{\mathcal{U}}, \mathbf{d}^{\mathcal{V}})$ and matching $M_T$ on it.

---

Since each pairing of each half-edge is done uniformly at random, the graph obtained at the end of the algorithm has indeed the law of a bipartite configuration model. Moreover, it is easy to see that M corresponds to the matching constructed by GREEDY MATCHING on $\mathbf{CM}(\mathbf{d}^{\mathcal{U}}, \mathbf{d}^{\mathcal{V}})$.

## 3.2 Differential Equation Method - Stochastic Approximation

As mentioned above, several quantities are going to be tracked through time: for all $k \in \{0, \ldots, T\}$ and all $i \geq 0$, we define:

- $F_i(k)$ as the number of vertices $u \in \mathcal{U}$ that are not yet matched at the end of step $k$ and whose remaining degree is $i$, meaning that $d_u - i$ of their initial half-edges have been paired. We will refer them to as *free* vertices.
- $M_i(k)$ as the number of vertices $u \in \mathcal{U}$ already matched at the end of step $k$ and whose remaining degree is $i$. We will refer them to as *marked* vertices.

Notice that for all $0 \leq k \leq T$, the sum $F_i(k) + M_i(k)$ corresponds to the total number of vertices of $\mathcal{U}$ with remaining degree $i$ at the end of step $k$. We also define

- $\widehat{F}(k) := \sum_{i \geq 0} i F_i(k)$ is the number of available half-edges attached to free vertices at the end of step $k$,

---
[5] Without loss of generality, we assume that the additional extra vertex is always on the $\mathcal{V}$ side.

- $\widehat{M}(k) := \sum_{i \geq 0} i M_i(k)$ is the number of available half-edges attached to marked vertices at the end of step $k$.

We are going to study the evolution of these quantities along with the one of GREEDY. A major ingredient of the proof is to show that $F_i(k)$ and $M_i(k)$ closely follow the solutions of some ODE. This is the so-called *differential equation method* [Wormald, 1995], stated in Appendix C. For instance, it can easily be seen that $\widehat{F}(k) + \widehat{M}(k)$ closely follows the function $t \mapsto \mu_{\mathcal{U}} - t\mu_{\mathcal{V}}$ on $(0, \mu_{\mathcal{U}}/\mu_{\mathcal{V}})$ in the following sense.

**Lemma 1.** *For every $\varepsilon > 0$, and for all $0 \leq k \leq T$,*

$$\left| \frac{\widehat{F}(k) + \widehat{M}(k)}{N} - \left( \mu_{\mathcal{U}} - \frac{k}{N}\mu_{\mathcal{V}} \right) \right) \right| \leq \varepsilon.$$

*with probability at least $1 - \exp\left( -\frac{N\epsilon^2}{2\sigma_U^2} \right) + \exp\left( -\frac{T\epsilon^2}{2\sigma_V^2} \right)$.*

We now turn to each individual quantity $F_i$ (resp. $M_i$). We can prove a similar result, yet the limit function is not explicit (unlike for the matching size as in Theorem 1 statement). The following Lemma 2 states that the discrete sequences of (free and marked) half-edges are closely related to the solutions of some system of differential equations.

Before stating it, we first introduce, for any sequence of non-negative numbers $(x_\ell)_{\ell \geq 0}$ and $(y_\ell)_{\ell \geq 0}$ such that $0 < \sum_\ell \ell(x_\ell + y_\ell) < \infty$, every $i \geq 0$, the following mappings

$$\Phi_i(x_0, x_1, \ldots, y_0, y_1, \ldots) := \frac{-i\mu_{\mathcal{V}}x_i + (i+1)\mu_{\mathcal{V}}x_{i+1} - h\left( \frac{\sum_{\ell \geq 0} \ell y_\ell}{\sum_{\ell \geq 0} \ell(x_\ell + y_\ell)} \right)(i+1)x_{i+1}}{\sum_{\ell \geq 0} \ell(x_\ell + y_\ell)} \quad (5)$$

and

$$\Psi_i(x_0, x_1, \ldots, y_0, y_1, \ldots) := \frac{-i\mu_{\mathcal{V}}y_i + (i+1)\mu_{\mathcal{V}}y_{i+1} + h\left( \frac{\sum_{\ell \geq 0} \ell y_\ell}{\sum_{\ell \geq 0} \ell(x_\ell + y_\ell)} \right)(i+1)x_{i+1}}{\sum_{\ell \geq 0} \ell(x_\ell + y_\ell)},$$

where $h$ is the following function, well-defined on $[0, 1]$,

$$h(s) = \frac{1 - \phi_{\mathcal{V}}(s)}{1 - s}.$$

**Lemma 2.** *With probability $1 - \zeta N \exp(-\xi N^{c/2})$, there are at most $N^c$ quantities $F_i$ and $M_i$, and for all $0 \leq k \leq T$ and all $i \geq 0$*

$$\left| \frac{F_i(k)}{N} - f_i\left( \frac{k}{N} \right) \right| \leq \kappa N^{-2c} \quad and \quad \left| \frac{M_i(k)}{N} - m_i\left( \frac{k}{N} \right) \right| \leq \kappa N^{-2c},$$

*where $\zeta, \kappa$ depend only on the (first two) moments of $\pi_{\mathcal{V}}$ and $\pi_{\mathcal{U}}$ and $c = 1/20$.*

*The continuous mappings $f_i$ and $m_i$ are solutions of the system of differential equations on $[0, \mu_{\mathcal{U}}/\mu_{\mathcal{V}})$*

$$\begin{aligned} \frac{df_i}{dt} &= \Phi_i(f_0, f_1, \ldots, m_0, m_1, \ldots), \\ \frac{dm_i}{dt} &= \Psi_i(f_0, f_1, \ldots, m_0, m_1, \ldots), \\ f_i(0) &= \pi_{\mathcal{U}}(i), \\ m_i(0) &= 0. \end{aligned} \quad (6)$$

This system is well defined as stated by the following Lemma 3.

**Lemma 3.** *The system (6) has a unique solution which is well-defined on $[0, \mu_{\mathcal{U}}/\mu_{\mathcal{V}})$. More precisely, denoting by $f$ and $m$ the generating series of the sequences $(f_i)_{i \geq 0}$ and $(m_i)_{i \geq 0}$,*

$$f(t, s) = \sum_{i \geq 0} f_i(t)s^i \quad and \quad m(t, s) = \sum_{i \geq 0} m_i(t)s^i,$$

*it holds that:*

$$f\left( \frac{\mu_{\mathcal{U}}}{\mu_{\mathcal{V}}} \left( 1 - e^{-\mu_{\mathcal{V}}t} \right), s \right) = \phi_{\mathcal{U}}\left( (s-1)e^{-\mu_{\mathcal{V}}t} + 1 - F(t) \right), \quad (7)$$

*and*

$$m\left(\frac{\mu_{\mathcal{U}}}{\mu_{\mathcal{V}}}\left(1 - e^{-\mu_{\mathcal{V}}t}\right), s\right) = \int_0^t F'(u)\phi_{\mathcal{U}}'\left((s-1)e^{-\mu_{\mathcal{V}}u} + 1 - F(u)\right)\mathrm{d}u.$$

*where $F$ is a solution of the following ODE*

$$\frac{\frac{1}{\mu_{\mathcal{U}}}\phi_{\mathcal{U}}'(1 - F(t))}{1 - \phi_{\mathcal{V}}\left(1 - \frac{1}{\mu_{\mathcal{U}}}\phi_{\mathcal{U}}'(1 - F(t))\right)}F'(t) = e^{-\mu_{\mathcal{V}}t}.$$

### 3.3 Aggregating solutions to compute GREEDY performances

To get Theorem 1, notice that the number of vertices matched by GREEDY is $N$ minus the number of free vertices remaining at the end, which is approximately equal to $Nf(\frac{\mu_{\mathcal{U}}}{\mu_{\mathcal{V}}}, 1)$ by definition of $f$ and because of Lemma 2. This corresponds to $t = +\infty$ in Equation (7), thus the performance of GREEDY is, with arbitrarily high probability, arbitrarily close to

$$N(1 - \phi_{\mathcal{U}}(1 - F(+\infty)))$$

The statement of Theorem 1 just follows from a simple final change of variable.

## Conclusion

We studied theoretical performances of GREEDY algorithm on matching problems with different underlying structures. Those precise results are quite interesting and raise many questions, especially since GREEDY actually outperforms RANKING in many different situations (in theory for 2-regular graphs, but empirical evidence indicates that this happens more generically).

Our approach has also successfully been used to unveil some questions on the comparison between different possible models. But more general questions are still open; for instance, assuming that the expected degree is fixed, which situation is the more favorable to GREEDY and online algorithm: small or high variance, or more generally this distribution $\pi_{\mathcal{U}}$ or an alternative one $\pi_{\mathcal{U}}'$ ? The obvious technique would be to compare the solution of the different associated ODE's. Similarly, the questions of stability/robustness of the solution to variation in the distribution $\pi_{\mathcal{U}}$ and $\pi_{\mathcal{V}}$ are quite challenging and left for future work.

We believe online matching will become an important problem for the machine learning community in the future. Each year, the complexity of the underlying graphs increases and we are considering adding features to the model in future work (such as random variables on the edges, modeling the interest for a consumer for a given product), or connection modeled via some Kernel between vertices features (say, if users and products/campaigns are embedded in the same space). In this context, machine learning tools will certainly be needed to tackle the problem.

## Acknowledgments and Disclosure of Funding

V. Perchet acknowledges support from the ANR under grant number #ANR-19-CE23-0026 as well as the support grant as part of the Investissement d'avenir project, reference LabEx Ecodec/ANR-11-LABX-0047. Nathan Noiry also acknowledges support from the Telecom Paris DSAIDIS chair and from the ANR ProGraM (ANR-19-CE40-0025). Flore Sentenac is supported by IP PARIS' PhD Funding.

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
