# A  General version of the result

## A.1  The fixed capacity matching problem

We now investigate the case where vertices $u \in \mathcal{U}$ have capacities, which means that they can be matched to several vertices $v \in \mathcal{V}$. Precisely, if the capacity of $u$ is denoted by $\omega_u$, then this vertex can be matched to at most $\omega_u$ vertices in $\mathcal{V}$ (but as before, half-edges of $u$ are going to be paired with $d_u$ half-edges originating from $\mathcal{V}$). The graph is still constructed using the configuration model introduced in Section 2.1, i.e., the law of $d_u$ is $\pi_{\mathcal{U}}$ (and similarly, degrees of $v \in \mathcal{V}$ are i.i.d., with law $\pi_{\mathcal{V}}$).

For the moment, to simplify the analysis and the results statements, we are going to assume that all vertices $u \in \mathcal{U}$ have the same initial capacity $C \in \mathbb{N}$. We denote the random graph with capacities generated this way by $\mathbf{CM}(\mathbf{d}^{\mathcal{U}}, \mathbf{d}^{\mathcal{V}}, C)$

**Theorem 2.** *(Performances of* GREEDY *with fixed capacities)*

*Given $N \geq 1$ and $T = \frac{\mu_{\mathcal{U}}}{\mu_{\mathcal{V}}} N$, let $\mathrm{M}_T$ be the matching built by* GREEDY *on $\mathbf{CM}(\mathbf{d}^{\mathcal{U}}, \mathbf{d}^{\mathcal{V}}, C)$ then the following convergence in probability holds:*

$$\frac{|\mathrm{M}_T|}{CN} \xrightarrow[N \to +\infty]{\mathbf{P}} 1 - \sum_{k=0}^{C-1} \frac{1 - k/C}{k!} G(1)^k \phi_{\mathcal{U}}^{(k)} (1 - G(1)).$$

*where $G$ is the unique solution of the following ordinary differential equation*

$$G'(s) = \frac{1 - \phi_{\mathcal{V}} \left(1 - \frac{1}{\mu_{\mathcal{U}}} \Gamma_U(G(s))\right)}{\frac{\mu_{\mathcal{V}}}{\mu_{\mathcal{U}}} \Gamma_U(G(s))}.$$

*where*

$$\Gamma_U(g) = \phi_{\mathcal{U}}'(1 - g) + \sum_{k=1}^{C-1} \frac{g^k}{k!} \phi_{\mathcal{U}}^{(k+1)}(1 - g)$$

*Moreover, for any $s \in [0, 1]$, if $M_T(s)$ is the matching obtained by* GREEDY *after seeing a proportion $s$ of vertices of $\mathcal{V}$, then*

$$\frac{|\mathrm{M}_T(s)|}{CN} \xrightarrow[N \to +\infty]{\mathbf{P}} 1 - \sum_{k=0}^{C-1} \frac{1 - k/C}{k!} G(s)^k \phi_{\mathcal{U}}^{(k)} (1 - G(s)).$$

The proof of Theorem 2, in Appendix E, has three major differences with the one of Theorem 1:

1.  The first one is that more quantities must be tracked, not just the number of vertices with remaining free half-edges, but the number of such vertices for each possible value of remaining capacity; the total number of equations is roughly speaking multiplied by a factor $(C+1)/2$ (since only $F_i(k)$ are affected by the capacities and not $M_i(k)$). We will therefore denote in the remaining by $F_i^{(c)}(k)$ the number of vertices with $i$ remaining half-edges to be paired and with current capacity equal to $c$ (those vertices can still be matched to $c$ different vertices $v \in \mathcal{V}$).

2.  The second major difference lies in the resolution of the system of differential equations. The solution was rather direct without capacities (i.e., $c = 1$). Unfortunately, the evolution of $F_i^{(c)}$ strongly depends on $F_i^{(c+1)}$. As a consequence, the trick is to solve this system by induction, starting from $c = C$ (this solution is almost identical to that of the case with no capacities) and then to inject this solution in the PDEs defining $F_i^{(C-1)}$ so on so forth. Indeed, the fluid limits of $\sum_i F_i(c)$ and $\sum_i M_i$, that we denote respectively be $f^{(c)}$ and $m$ satisfy the following coupled equations (up to some time change $\theta(t)$ and where $H(t) = h(q(t))$ for some function $q(\cdot)$ introduced in the proof):

    $$\partial_t f^{(c)}(\theta(t), s) = [-\mu_{\mathcal{V}} s + \mu_{\mathcal{V}} - H(t)] \partial_s f^{(c)}(\theta(t), s) + H(t) \partial_s f^{(c+1)}(\theta(t), s),$$

    and

    $$\partial_t m(\theta(t), s) = [-\mu_{\mathcal{V}} s + \mu_{\mathcal{V}}] \partial_s m(\theta(t), s) + H(t) \partial_s f^{(1)}(\theta(t), s).$$

3. Finally, the third main difference is how the performances of GREEDY are defined. The upper bound is obviously to create the minimum between $CN$ and $T$ matches (where $T$ is the number of vertices in $\mathcal{V}$). Anyway, those two numbers are within a constant multiplicative factor (recall that $T = \frac{\mu_{\mathcal{U}}}{\mu_{\mathcal{V}}}N$ for a valid configuration model), hence we arbitrarily chose to normalize GREEDY performances by $CN$. As a consequence, the (normalized) performances of GREEDY now rewrite as

$$\frac{\sum_{i\geq 0}\left(M_i(T) + \sum_{c=1}^{C}(1 - \frac{c}{C})F_i^{(c)}(T)\right)}{N},$$

where $M_i(k)$ still denotes the number of marked vertices, i.e., those whose capacities have been depleted before step $k$ with $i$ remaining half-edges to be paired.

## A.2 General case, online matching with capacities

In the general case, we no longer assume that all vertices $u \in \mathcal{U}$ have the same initial capacities, but $\omega_u$ can be equal to any value in $\mathbb{N}$ (yet this capacity is independent of the degree). Notice however that the capacities $\omega_u$ of vertices could be capped at their degrees $d_u$ (since they would never be depleted otherwise). As a consequence, capacities can be assumed to be bounded by $C < N^\beta$ for some $\beta < 1$ since the maximal degree is also smaller than $N^\beta$ with arbitrarily high probability.

We therefore denote by $p_c \in [0,1]$ the fraction of vertices of $\mathcal{U}$ whose initial capacity is exactly $c \in [1, C]$. Notice, we do not need to assume that capacities are drawn i.i.d. accordingly to some distribution, our results hold for any values $(p_c)_c$. We denote by $\mathbf{CM}(\mathbf{d}^{\mathcal{U}}, \mathbf{d}^{\mathcal{V}}, \mathbf{p})$ the random graph with capacities generated.

Quite interestingly, the techniques are exactly the same as in the previous case: we consider the exact same system of differential equations; the only differences are the initial conditions. Similarly, the maximal matching size is no longer $NC$ but $N\mathbb{E}_{\mathbf{p}}[c] := N\sum_c cp_c$. We also denote the cdf of the empirical distribution $p_c$ by $P_c := \sum_{k\leq c} p_c$

**Theorem 3.** *(Performances of GREEDY with different capacities)*

*Given $N \geq 1$ and $T = \frac{\mu_{\mathcal{U}}}{\mu_{\mathcal{V}}}N$, let $\mathrm{M}_T$ be the matching built by GREEDY on $\mathbf{CM}(\mathbf{d}^U, \mathbf{d}^V, \mathbf{p})$ then the following convergence holds in probability:*

$$\frac{|\mathrm{M}_T|}{N\mathbb{E}_{\mathbf{p}}[c]} \xrightarrow[N\to+\infty]{\mathbf{P}} 1 - \sum_{k=0}^{C-1}\frac{\sum_{c=1}^{C}cp_{c+k}}{\mathbb{E}_{\mathbf{p}}[c]}\frac{1}{k!}G(1)^k\phi^{(k)}\left(1 - G(1)\right).$$

*where $G$ is the unique solution of the following ordinary differential equation*

$$G'(s) = \frac{1 - \phi_{\mathcal{V}}\left(1 - \frac{1}{\mu_{\mathcal{U}}}\Gamma_U^{\mathbf{P}}(G(s))\right)}{\frac{\mu_{\mathcal{V}}}{\mu_{\mathcal{U}}}\Gamma_U^{\mathbf{P}}(G(s))}.$$

*with*

$$\Gamma_U^{\mathbf{P}}(g)) = \phi_U'(1 - g) + \sum_{k=1}^{C-1}\left(\frac{(1 - P_k)g^k}{k!}\phi_{\mathcal{U}}^{(k+1)}(1 - g)\right).$$

*Moreover, for any $s \in [0,1]$, if $M_T(s)$ is the matching obtained by GREEDY after seeing a proportion $s$ of vertices of $\mathcal{V}$, then*

$$\frac{|\mathrm{M}_T(s)|}{N\mathbb{E}_{\mathbf{p}}[c]} \xrightarrow[N\to+\infty]{\mathbf{P}} 1 - \sum_{k=0}^{C-1}\frac{\sum_{c=1}^{C}cp_{c+k}}{\mathbb{E}_{\mathbf{p}}[c]}\frac{1}{k!}G(s)^k\phi^{(k)}\left(1 - G(s)\right).$$

As mentioned before, the proof (delayed to Appendix F) is rather similar to the previous one; the major difference is that the change of initial condition of the system of PDE makes it a bit more complicated to solve (hence the more intricate formulation of the result).

# B Additional Numerical Experiments

## B.1 Further comparisons between the theoretical result and simulations

We provide in Figure 3 a comparison between the score predicted by the numerical solutions of the ODE (the functions $1 - \phi_{\mathcal{U}}(1 - G(s))$) for 4-regular graphs and the simulated performance of GREEDY for various values of $N$. As expected, the deviations of the simulated trajectories remain within $\mathcal{O}(\sqrt{N})$ of the expected theoretical trajectory. Figure 4 illustrates the same comparison on an Erdős-Rényi graph whose expected degree equals 4.

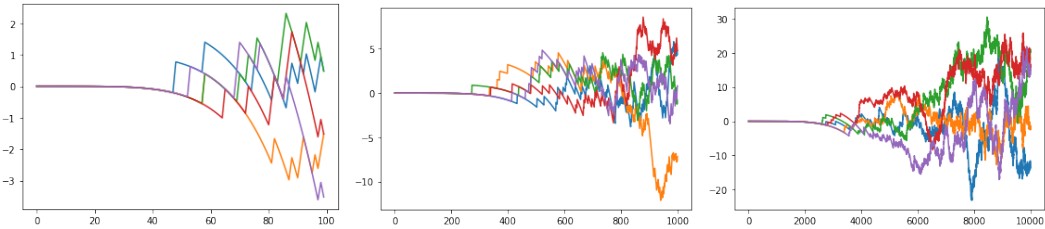

Figure 3: Difference between the theoretical performances and simulated performances of the GREEDY algorithm on the $d$-regular graph ($d = 4$) on 5 independent runs, with $N = 100, 1000, 10000$.

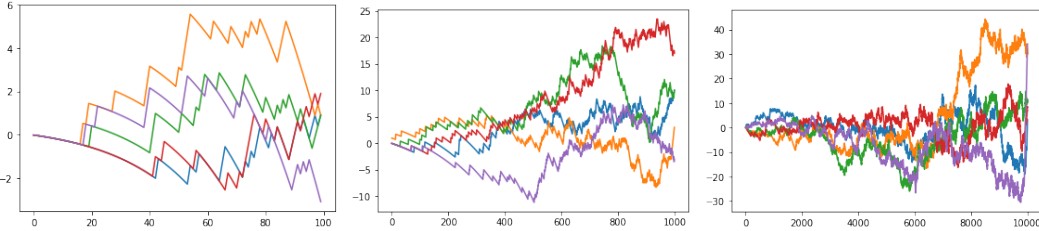

Figure 4: Difference between the theoretical value 2 and simulated performances of the GREEDY algorithm on the Erdős-Rényi graph, $c = 4$, on 5 independent runs, with $N = 100, 1000, 10000$.

In Figure 5, we plot the theoretical performance of the GREEDY algorithm along with its experimental performance on the $d$-regular graph for various values of $d$. We also plot the competitive ratio of GREEDY predicted by the ODE as a function of $d$. As expected, the score increases with $d$ (as the problem becomes simpler and simpler for GREEDY when $d \geq 2$).

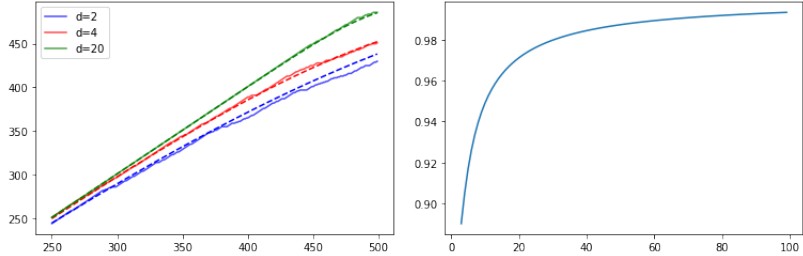

Figure 5: On the left, the expected theoretical performance of the GREEDY algorithm (dashed line) along with the simulated performance (full line) for various values of $d$. On the right, the expected competitive ratio of GREEDY on the $d$-regular graph as a function of $d$.

## B.2 GREEDY vs RANKING

We further illustrate in this section the quite surprising fact that, in some configuration models, GREEDY actually outperforms RANKING.

In the adversarial configuration, it is known that the competitive ratio of RANKING is $1 - \frac{1}{e}$ which is bigger than the one of GREEDY, equal to $1/2$, see [Mehta, 2012]. In the following figures, we also plot the performances of two other "algorithms" SMALLEST and HIGHEST, for the sake of comparison; indeed, those are not admissible algorithms as they use the (future) knowledge of the number of half-edges of each vertex $u \in \mathcal{U}$.

More precisely, SMALLEST matches a vertex $v_k \in \mathcal{V}$ to the vertex $u \in \mathcal{U}$ with the smallest number of remaining half-edges (under the constraints obviously that $(u, v_k) \in \mathcal{E}$). As a consequence SMALLEST could be seen as an upper limit for an online algorithm.

HIGHEST does the opposite: it matches $v_k$ to the vertex $u \in \mathcal{U}$ with the highest remaining number of half-edges. So HIGHEST should serve as a lower bound/sanity check for any online algorithm.

In Figure 6, the performances of those 4 matching "algorithms" (again SMALLEST and HIGHEST are not admissible as they use extra knowledge) are illustrated on configuration models with $d = 2, 4, 10$ and $20$.

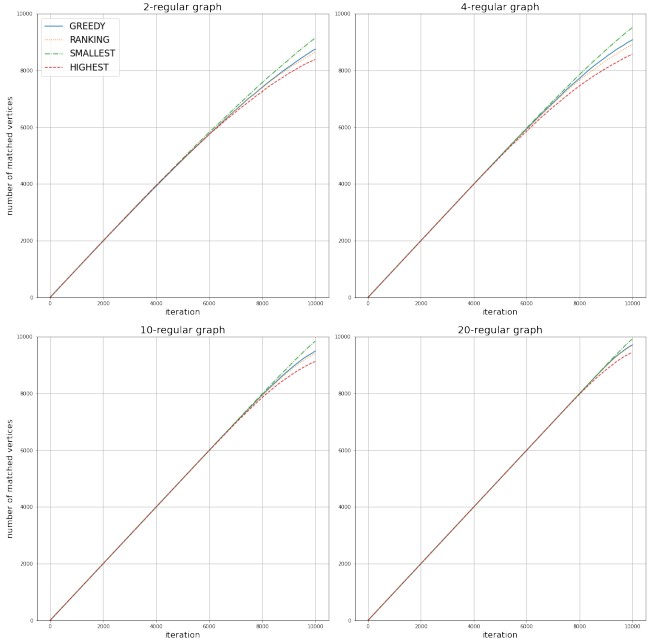

Figure 6: GREEDY outperforms RANKING in $d$-regular graphs

As mentioned before, GREEDY surprisingly outperforms RANKING in some configuration models, with a relative performance that decreases with $d$ (which is rather natural on the other hand, since the relative performance of HIGHEST and SMALLEST also decreases).

Figure 6 also illustrates the different time steps at which algorithms fail to match new vertices $v_k$ (because all the $u$ they are paired with are already matched with another vertex $v_j$ for some $j < k$). This happens later and later as $d$ increases (as expected), at around half the horizon for $d = 2$ and roughly $82\%$ with $d = 20$.

On the other hand, RANKING and GREEDY have the same performance on Erdos-Renyi graphs, which is a consequence of the memory-less property of those graphs, i.e. the probability of creating a match at each iteration depends only on the number of matched vertices, as shown in Figure 7.

In Figure 8, we plot the relative performance of RANKING and GREEDY on bi-degrees graph, where half the vertices have degree $x$, the other half $2x$. The plots illustrate that the best algorithm is not always teh best one depending on the value of $x$.

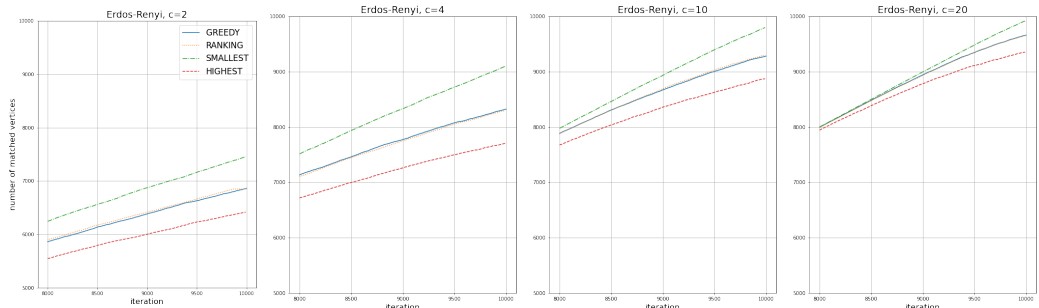

Figure 7: Experimental performances of GREEDY vs. RANKING on Erdos-Renyi graphs

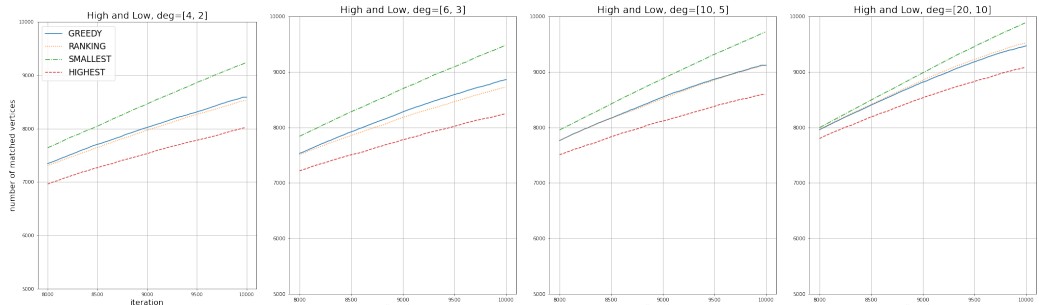

Figure 8: Experimental performances of GREEDY vs. RANKING on bi-degrees graphs

### B.3 A few vertices with high capacity vs many vertices with low capacity

In this section, we investigate how nodes' capacities affect GREEDY's expected performance. The baseline is its performance on a random graph where all vertices have capacity 1 and the vertices degrees in $\mathcal{U}$ and $\mathcal{V}$ follow the distributions $\pi_{\mathcal{U}}$ and $\pi_{\mathcal{V}}$. The comparison graph with capacity $C$ has $|\mathcal{U}|/C$ "in-place" vertices, each with a capacity $C$, and their degrees follows the modified distribution $\tilde{\pi}_C^U$ where $\tilde{\pi}_C^U(x = k) = \pi_{\mathcal{U}}(x = k/C)$. Informally, the graph with capacity $C$ is built from the baseline graph by merging $C$ vertices of equal degree $d$ into a single vertex of degree $dC$.

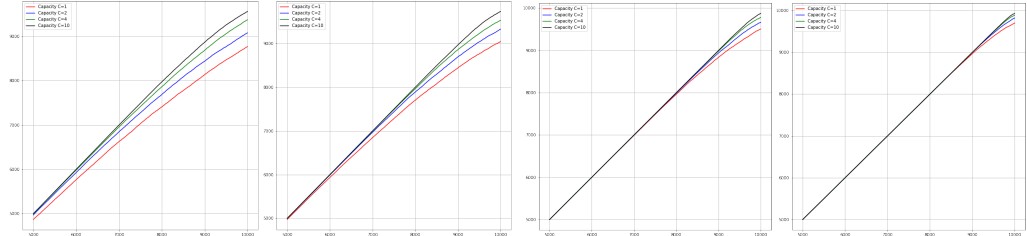

Figure 9: GREEDY performs better in high capacity graphs in $d$-regular graphs, from left to right $d = [2, 4, 10, 20]$

The results of the simulation illustrate that the GREEDY performs better on graphs with vertices of high capacity.

## C Stochastic approximation & Differential equation method

The following theorem is an improved version of Wormald's Theorem [Enriquez et al., 2019].

**Theorem 4.** *Let $a > 0$. For all $N \geq 1$ and all $1 \leq k \leq N^a$, let $Y_k(i) = Y_k^{(N)}(i)$ be a Markov chain with respect to a filtration $\{\mathcal{F}_i\}_{i \geq 1}$. Suppose that, for all $k \geq 1$, there exists a function $f_k$ such that:*

- $Y_k(0)/N = z_k(0)$;

- $|Y_k(i+1) - Y_k(i)| \leq N^\beta$;

- $\left| \mathbf{E} \left[ Y_k(i+1) - Y_k(i) \,\middle|\, \mathcal{F}_i \right] - f_k \left( \frac{i}{N}, \frac{(Y_k(i))_{1 \leq k \leq N^a}}{N} \right) \right| \leq cN^{-\lambda}$, *for some constant $c > 0$*

*where $0 < \beta < 1/2$, $\lambda > 0$. Suppose that the following infinite system of differential equations with initial conditions $(z_k(0))_{k \geq 1}$ has a unique solution $(z_k)_{k \geq 1}$:*

$$\forall k \geq 1, \quad z'_k(t) = f_k(t, (z_k(t))_{k \geq 1}).$$

*Then, for all $k \geq 1$, $Y_k(\lfloor tN \rfloor)/N$ converges in probability towards $z_k$ for the topology of uniform convergence.*

*More precisely, for every $1 < \varepsilon < \frac{1-\beta}{\beta}$, for every $\frac{(1+\varepsilon)\beta}{2} < \alpha < \varepsilon\beta$ and for every $0 \leq i \leq \frac{N}{\omega}$ where $\omega = N^{(1+\varepsilon)\beta}$, it holds that*

$$\mathbb{P}\left( |Y(i\omega) - z(\tfrac{i\omega}{N})N| \leq i\left( N^{\alpha+\beta} + cN^{(1+\varepsilon)\beta-\lambda} + N^{2(1+\varepsilon)\beta-2} \right) \right) \leq i \exp\left( -\frac{N^{2\alpha-(1+\varepsilon)\beta}}{2} \right)$$

## D  Proofs of technical steps of Theorem 1

### D.1  Proof of Lemma 1

It is an application of (maximal) Hoeffding-Azuma inequality since, for every $0 \leq k \leq M - 1$,

$$\mathbf{E}\left[ \left( \widehat{F}(k+1) + \widehat{M}(k+1) \right) - \left( \widehat{F}(k) + \widehat{M}(k) \right) \,\middle|\, \mathcal{F}_k \right] = -\mathbf{E}\left[ d_k^{\mathcal{V}} \right] = -\mu_{\mathcal{V}}.$$

### D.2  Proof of Lemma 2

Since $\pi_{\mathcal{V}}$ is $\sigma_{\mathcal{V}}$ subGaussian, then for any $\beta > 0$,

$$\mathbf{P}\left( \exists i \in \{1, \ldots, T\}, \, d_i^{\mathcal{V}} \geq \mu_{\mathcal{V}} + N^\beta \right) \leq T \exp\left(-\frac{N^{2\beta}}{2\sigma_{\mathcal{V}}^2}\right).$$

In particular, for some $\beta < 1/2$ to be chosen later on, if $\mu_{\mathcal{V}} \leq N^\beta/2$, then all degrees are smaller than $N^\beta$ with probability at least $1 - T \exp\left( -\frac{N^{2\beta}}{8\sigma_{\mathcal{V}}^2} \right)$; from now on, we will place ourselves on that event.

We also denote by $(\mathcal{F}_k)_{0 \leq k \leq M}$ the natural filtration associated to the GREEDY MATCHING algorithm. In order to apply Theorem 4, it remains to control for every $i \geq 0$ and $0 \leq k \leq M - 1$,

$$\left| \mathbf{E}\left[ F_i(k+1) - F_i(k) \,\middle|\, \mathcal{F}_k \right] - \Phi_i\left( F_0\left(\frac{k}{N}\right), F_1\left(\frac{k}{N}\right), \ldots, M_0\left(\frac{k}{N}\right), M_1\left(\frac{k}{N}\right), \ldots \right) \right|$$

and

$$\left| \mathbf{E}\left[ M_i(k+1) - M_i(k) \,\middle|\, \mathcal{F}_k \right] - \Psi_i\left( F_0\left(\frac{k}{N}\right), F_1\left(\frac{k}{N}\right), \ldots, M_0\left(\frac{k}{N}\right), M_1\left(\frac{k}{N}\right), \ldots \right) \right|$$

Let $0 \leq k \leq T - \frac{2N^\gamma}{\mu^{\mathcal{V}}}$, with $\gamma > 1/2$ some parameter to be fixed later, so that, according to Lemma 1, with probability at least $1 - \exp(-\frac{N^{2\gamma-1}}{2\sigma_{\mathcal{U}}^2}) - \exp(-\frac{\mu_{\mathcal{U}} N^{2\gamma-1}}{2\mu_{\mathcal{V}}\sigma_{\mathcal{V}}^2})$ it holds that $\widehat{F}(k) + \widehat{M}(k) \geq N^\gamma$.

Recall that, in the $k$-th step of the algorithm, half-edges of the $k$-th vertex of $V_N$ are ordered uniformly at random: $(e_i^k)_i$ for $i = 1, \ldots, d_k^{\mathcal{V}}$. Then, each of these half-edges is sequentially paired uniformly at random with half-edges of $d^{\mathcal{V}}$ that are not yet paired. Let $u_i^k$ be the vertex to which $e_i^k$ is paired and let $I_k$ be the first integer $i$ such that $u_i^k$ belongs to the free vertices of $\mathcal{U}$ at time $k$, that is to the vertices that are not yet matched. If such an integer does not exist, that is when all $u_i^k$ are already

matched, we set $I_k = +\infty$. As a consequence, we aim at estimating $\mathbf{P}\left(I_k = i \mid \mathcal{F}_k\right)$ for the different admissible values, where this probability has the following explicit definition

$$\mathbf{P}\left(I_k = i \mid \mathcal{F}_k\right) = \frac{\widehat{M}(k)}{\widehat{F}(k) + \widehat{M}(k)} \frac{\widehat{M}(k) - 1}{\widehat{F}(k) + \widehat{M}(k) - 1} \cdots \frac{\widehat{M}(k) - (i-2)}{\widehat{F}(k) + \widehat{M}(k) - (i-2)} \frac{\widehat{F}(k)}{\widehat{F}(k) + \widehat{M}(k) - (i-1)}$$

$$= \frac{\widehat{M}(k)!}{(\widehat{M}(k) - (i-1))!} \frac{(\widehat{F}(k) + \widehat{M}(k) - i)!}{(\widehat{F}(k) + \widehat{M}(k))!} \widehat{F}(k)$$

First, assume that $\widehat{M}(k) \geq 2N^\theta$ for some parameter $\theta > 2\beta$ to be chosen later, so that those probabilities are all strictly positive. Using Stirling approximation formula, we get that, with $p(k) = \frac{\widehat{M}(k)}{\widehat{F}(k) + \widehat{M}(k)}$ and for any $i$,

$$0 \geq \frac{\mathbf{P}\left(I_k = i \mid \mathcal{F}_k\right) - (1 - p(k))^{i-1} p(k)}{(1 - p(k))^{i-1} p(k)} \geq -2\frac{N^{2\beta}}{N^\theta} - \frac{N^\beta}{N^\gamma}$$

Second, assume that $\widehat{M}(k) < 2N^\theta$ for some $\theta > \beta$. This immediately implies that, for $i$,

$$0 \geq \mathbf{P}\left(I_k = i \mid \mathcal{F}_k\right) - (1 - p(k))^{i-1} p(k) \geq -2\frac{N^\theta}{N^\gamma}$$

Similar inequalities holds for $\mathbf{P}(I_k = +\infty \mid \mathcal{F}_k)$, except that it is approximately equal to $\mathbf{E}\left[(1 - p(k))^{d_k^\mathcal{V}}\right] = \phi_\mathcal{V}(1 - p(k))$.

It remains to control the evolution of the processes $F_i(k)$ and $M_i(k)$. Notice that, by their very definition, on the event $I_k = x$ for some $1 \leq x \leq d_k^\mathcal{V}$, the following happens:

1. The first $x - 1$ half-edges $e_k^1, \ldots, e_k^x$ are paired uniformly at random with marked half-edges of $\mathcal{U}$. If the corresponding vertex has a remaining degree equal to $i$, then $M_i$ decreases by one, and $M_{i-1}$ increases by one.

2. The $x$-th half-edge $e_k^x$ is paired uniformly at random with free half-edge of $\mathcal{U}$. If the corresponding vertex has a remaining degree $i$, then $F_i$ decreases by one, and $M_{i-1}$ increases by one.

3. The $d_k^\mathcal{V} - x$ remaining half-edges $e_k^{x+1}, \ldots, e_k^{d_k^\mathcal{V}}$ are paired uniformly at random with half-edges of $\mathcal{U}$. If the corresponding vertex is free with remaining degree $i$, then $F_i$ decreases by one and $F_{i-1}$ increases by one. Otherwise, if the corresponding vertex is marked with remaining degree $i$, then $M_i$ decreases by one and $M_{i-1}$ increases by one.

Notice that, after the pairing of each half-edges, the quantity $\widehat{F}(k)$ (resp. $\widehat{M}(k)$) may decrease (resp. increase) by one. Therefore, working on the event where $d_k^\mathcal{V} \leq N^\beta$, we deduce that $\widehat{F}$ and $\widehat{M}$ are affected by an additive term of order at most $N^\beta$. The same argument holds on $F_i$ and $M_i$.

All of these considerations imply that

$$\left| \mathbf{E}[F_i(k+1) - F_i(k) \mid \mathcal{F}_k, I_t = x] - \left( -\frac{iF_i(k)}{\widehat{F}(k)} + (\mu_\mathcal{V} - x)\left( -\frac{iF_i(k)}{\widehat{F}(k) + \widehat{M}(k)} + \frac{(i+1)F_{i+1}(k)}{\widehat{F}(k) + \widehat{M}(k)} \right) \right) \right|$$

$$\leq 2\sigma_{\mu_\mathcal{V}}^2 \frac{N^\beta}{N^\gamma}$$

and similarly

$$\left| \mathbf{E}[M_i(k+1) - M_i(k) \mid \mathcal{F}_k, I_t = x] - \left( (x-1)\left( -\frac{iM_i}{\widehat{M}} + \frac{(i+1)M_{i+1}}{\widehat{M}} \right) + \frac{(i+1)F_{i+1}}{\widehat{F}} \right. \right.$$

$$\left. \left. + (\mu_\mathcal{V} - x)\left( -\frac{iM_i}{\widehat{F} + \widehat{M}} + \frac{(i+1)M_{i+1}}{\widehat{F} + \widehat{M}} \right) \right) \right|$$

$$\leq 2\sigma_\mu^2 \frac{N^\beta}{N^\gamma} + 2\sum_{j=1}^{x-1} \frac{j}{\widehat{M}(k) - j}$$

Finally, the case $I_t = +\infty$ is handled similarly, as by definition

$$\mathbf{E}[F_i(k+1) - F_i(k) \,|\, \mathcal{F}_k, \, I_t = +\infty] = 0.$$

and the following also holds also holds:

$$\left| \mathbf{E}[M_i(k+1) - M_i(k) \,|\, \mathcal{F}_k, \, I_t = +\infty] - \mu_{\mathcal{V}}\left(-\frac{iM_i}{\widehat{M}} + \frac{(i+1)M_{i+1}}{\widehat{M}}\right) \right| \le 2\sigma_\mu^2 \frac{N^\beta}{N^\gamma} + 2\sum_{j=1}^{d_k^{\mathcal{V}}-1} \frac{j}{\widehat{M}(k) - j}.$$

It remains to compute the expected variation in $F_i(k)$ and $M_i(k)$. It is a bit simpler for the former, but still, to lighten the notations, we write $p = p_k$ and $q = q_k$ in the following computation.

$$\mathbf{E}\left[F_i(k+1) - F_i(k) \,\big|\, \mathcal{F}_k\right]$$

$$= \mathbf{E}_{d_k^{\mathcal{V}} \sim \pi_{\mathcal{V}}}\left[\sum_{x=1}^{d_k^{\mathcal{V}}} q^{x-1}\left(-\frac{iF_i}{\widehat{F}+\widehat{M}}\right) + p\sum_{x=1}^{d_k^{\mathcal{V}}} q^{x-1}(d_k^{\mathcal{V}} - x)\left(-\frac{iF_i}{\widehat{F}+\widehat{M}} + \frac{(i+1)F_{i+1}}{\widehat{F}+\widehat{M}}\right)\right] + \eta_N$$

$$= \frac{1}{\widehat{F}+\widehat{M}}\mathbf{E}_{d_k^{\mathcal{V}} \sim \pi_{\mathcal{V}}}\left[-iF_i\left(\sum_{i=1}^{d_k^{\mathcal{V}}} q^{x-1} + pd_k^{\mathcal{V}}\sum_{x=1}^{d_k^{\mathcal{V}}} q^{x-1} - p\sum_{x=1}^{d_k^{\mathcal{V}}} xq^{x-1}\right)\right.$$

$$\left. + (i+1)F_{i+1}\left(pd_k^{\mathcal{V}}\sum_{x=1}^{d_k^{\mathcal{V}}} q^{x-1} - p\sum_{x=1}^{d_k^{\mathcal{V}}} xq^{x-1}\right)\right] + \eta_N$$

$$= \frac{1}{\widehat{F}+\widehat{M}}\mathbf{E}_{d_k^{\mathcal{V}} \sim \pi_{\mathcal{V}}}\left[-iF_i\left(\frac{1-q^{d_k^{\mathcal{V}}}}{1-q} + d_k^{\mathcal{V}}(1-q^{d_k^{\mathcal{V}}}) - \frac{d_k^{\mathcal{V}} q^{d_k^{\mathcal{V}}+1} - (d_k^{\mathcal{V}}+1)q^{d_k^{\mathcal{V}}} + 1}{1-q}\right)\right.$$

$$\left. + (i+1)F_{i+1}\left(q(1-q^{d_k^{\mathcal{V}}}) - \frac{dq^{d_k^{\mathcal{V}}+1} - (d_k^{\mathcal{V}}+1)q^{d_k^{\mathcal{V}}} + 1}{1-q}\right)\right] + \eta_N$$

$$= \frac{1}{\widehat{F}+\widehat{M}}\mathbf{E}_{d_k^{\mathcal{V}} \sim \pi_{\mathcal{V}}}\left[-iF_i d_k^{\mathcal{V}} + (i+1)F_{i+1}\frac{q^{d_k^{\mathcal{V}}} - d_k^{\mathcal{V}} q + (d_k^{\mathcal{V}} - 1)}{1-q}\right] + \eta_N$$

$$= \frac{1}{\widehat{F}+\widehat{M}}\mathbf{E}_{d_k^{\mathcal{V}} \sim \pi_{\mathcal{V}}}\left[-iF_i + (i+1)d_k^{\mathcal{V}} F_{i+1} + (i+1)F_{i+1}\frac{1-q^{d_k^{\mathcal{V}}}}{1-q}\right] + \eta_N$$

$$= \frac{-i\mu_{\mathcal{V}} F_i + (i+1)\mu_{\mathcal{V}} F_{i+1} - (i+1)h(q)F_{i+1}}{\widehat{F}+\widehat{M}} + \eta_N,$$

which is exactly (5), up to error term $\eta_N$ that satisfies, if $\widehat{M}(k) < 2N^\theta$,

$$|\eta_N| \le 2\sigma_{\mu_{\mathcal{V}}}^2 \frac{N^\beta}{N^\gamma} + 2\mu_{\mathcal{V}}^2 \frac{N^\theta}{N^\gamma}$$

and, if $\widehat{M}(k) \ge 2N^\theta$,

$$|\eta_N| \le 3\sigma_{\mu_{\mathcal{V}}}^2 \frac{N^\beta}{N^\gamma} + 2\mu_{\mathcal{V}}^2 \frac{N^{2\beta}}{N^\theta} + \mu_{\mathcal{V}}^2 \frac{N^\beta}{N^\gamma}$$

Computations are quite similar for the difference in $M_i(k)$ and the error term still depends whether $\widehat{M}(k)$ is bigger, or smaller, than $2N^\theta$:

$$\mathbf{E}\big[M_i(k+1) - M_i(k) \,\big|\, \mathcal{F}_k\big]$$

$$= \mathbf{E}_{d_k^\mathcal{V} \sim \pi_\mathcal{V}}\left[\left(\frac{d_k^\mathcal{V}(1-p)^{d_k^\mathcal{V}}}{\widehat{M}} + \sum_{x=1}^{d_k^\mathcal{V}} pq^{x-1}\left(\frac{(x-1)}{\widehat{M}} + \frac{d_k^\mathcal{V} - x}{\widehat{M} + \widehat{F}}\right)\right)\big((i+1)M_{i+1} - iM_i\big)\right]$$

$$+ \mathbf{E}_{d_k^\mathcal{V} \sim \pi_\mathcal{V}}\left[\sum_{x=1}^{d_k^\mathcal{V}} pq^{x-1}\frac{(i+1)F_{i+1}}{\widehat{F}}\right] + \varepsilon_N$$

$$= \frac{1}{\widehat{M} + \widehat{F}}\mathbf{E}_{d_k^\mathcal{V} \sim \pi_\mathcal{V}}\left[\left(d_k^\mathcal{V}(1-p)^{d_k^\mathcal{V}-1} + \sum_{x=1}^{d_k^\mathcal{V}}\big(pq^{x-2}(x-1) + pq^{x-1}(d_k^\mathcal{V} - x)\big)\right)\big((i+1)M_{i+1} - iM_i\big)\right]$$

$$+ \frac{(i+1)h(q)F_{i+1}}{\widehat{M} + \widehat{F}} + \varepsilon_N$$

$$= \frac{\mu_\mathcal{V}((i+1)M_{i+1} - iM_i) + (i+1)h(q)F_{i+1}}{\widehat{F} + \widehat{M}} + \varepsilon_N,$$

where $\varepsilon_N$ satisfies, if $\widehat{M}(k) < 2N^\theta$,

$$|\varepsilon_n| \le 2\sigma_\mathcal{V}^2\frac{N^\theta}{N^\gamma} + 2\sigma_\mathcal{V}^2\frac{N^\beta}{N^\gamma} + 2\mu_\mathcal{V}^2\frac{N^\theta}{N^\gamma};$$

and, if $\widehat{M}(k) \ge 2N^\theta$, it satisfies

$$|\varepsilon_n| \le 3\sigma_\mathcal{V}^2\frac{N^\beta}{N^\gamma} + 2\mu_\mathcal{V}^2\frac{N^{2\beta}}{N^\theta} + \mu_\mathcal{V}^2\frac{N^\beta}{N^\gamma};$$

We used in the above computations (at the third equality) the following observation:

$$kq^{k-1} + \sum_{x=1}^{k} pq^{x-1}\left(\frac{x-1}{q} + k - x\right) = k$$

Summing error terms over all the $2N^\beta$ equations relating $F_i$ to $f_i$ and $M_j$ to $m_j$, the error terms coming from the differential equation method Theorem 4, and using the fact that $m$ is $\mu_\mathcal{U}$-Lipschitz, we get that the total error, defined by,

$$\text{Err} := \sup_{s \in [0,1]}\left|\frac{|\mathbf{M}_T(s)|}{N} - \big(1 - \phi_\mathcal{U}(1 - G(s))\big)\right|$$

satisfies

$$\text{Err} \le N^\beta N^{(1+\varepsilon)\beta}\big(N^{\alpha+\beta} + 4(\sigma_\mathcal{V}^2 + \mu_\mathcal{V}^2)N^{(1+\varepsilon)\beta - \gamma/2 + \beta} + N^{2(1+\varepsilon)\beta - 2}\big) + \mu_\mathcal{U}N^{(1+\varepsilon)\beta}$$

as soon as $\theta = \beta + \frac{\gamma}{2}$.

It remains to pick admissible values for the different parameters, such as the following ones (checking admissibility follows from immediate computations):

$$\beta = 1/20, \epsilon = 10, \gamma = 21/40, \theta = 25/80, \alpha.23/80$$

Those choices ensures that $\text{Err} = \mathcal{O}(N^{-1/20})$.

All those arguments hold with probability at least (summing all the bad event probabilities)

$$1 - T\exp(-\frac{N^{2\beta}}{2\sigma_V^2}) - \exp(-\frac{N^{2\gamma-1}}{2\sigma_\mathcal{U}^2}) - \exp(-\frac{\mu_\mathcal{U}N^{2\gamma-1}}{2\mu_\mathcal{V}\sigma_\mathcal{V}^2}) - 2N^\beta N^{(1+\varepsilon)\beta}\exp(-N^{2\alpha-(1+\varepsilon)\beta}) \ge 1 - \mathcal{O}(N\exp(-\zeta N^{1/40}))$$

where the equality holds because of the choice of parameters.

### D.3 Proof of Lemma 3

Notice that the functions $f$ and $m$ satisfy the following partial differential equations:

$$\partial_t f(t,s) = \frac{1}{\mu_{\mathcal{U}} - t\mu_{\mathcal{V}}} \left[-\mu_{\mathcal{V}}s + \mu_{\mathcal{V}} - h(q(t))\right]\partial_s f(t,s),$$

and

$$\partial_t m(t,s) = \frac{1}{\mu_{\mathcal{U}} - t\mu_{\mathcal{V}}} \left[-\mu_{\mathcal{V}}s + \mu_{\mathcal{V}}\right]\partial_s m(t,s) + h(q(t))\partial_s f(t,s),$$

where $q(t) = \partial_s f(t,1)/(\mu^{\mathcal{U}} - t\mu^{\mathcal{V}})$.

To solve these equations, we first perform a time change to get rid of the denominator. Let

$$\theta(t) = \frac{\mu_{\mathcal{U}}}{\mu_{\mathcal{V}}}\left(1 - e^{-\mu_{\mathcal{V}}t}\right)$$

so that $\theta'(t) = \mu_{\mathcal{U}} - \theta(t)\mu_{\mathcal{V}}$. In order to simplify notations, we set:

$$H(t) := h\left(q(\theta(t))\right).$$

Then, the new functions

$$g(t,s) := f(\theta(t),s) \qquad \text{and} \qquad o(t,s) := m(\theta(t),s)$$

satisfy the following PDEs:

$$\partial_t g(t,s) = \left[-\mu_{\mathcal{V}}s + \mu_{\mathcal{V}} - H(t)\right]\partial_s g(t,s), \tag{8}$$

and

$$\partial_t o(t,s) = \left[-\mu_{\mathcal{V}}s + \mu_{\mathcal{V}}\right]\partial_s o(t,s) + H(t)\partial_s g(t,s). \tag{9}$$

These two equations fall into the classical framework of *transport* differential equation and can be explicitly solved. We give the details for the reader's convenience.

**Solution of** (8). Let $s$ be a solution of the following ODE:

$$s'(t) = \mu_{\mathcal{V}}s(t) - \mu_{\mathcal{V}} + H(t). \tag{10}$$

Then, the function $g$ is constant along the curve $(t,s(t))$. Indeed:

$$\frac{\mathrm{d}}{\mathrm{d}t}g(t,s(t)) = \partial_t g(t,s) + s'(t)\partial_s g(t,s) = 0.$$

The differential equation (10) admits the following general solutions:

$$s_c(t) = \left[c + e^{-\mu_{\mathcal{V}}t} - 1 + \int_0^t e^{-\mu_{\mathcal{V}}u}H(u)\mathrm{d}u\right]e^{\mu_{\mathcal{V}}t}.$$

Therefore,

$$(t,s) = (t,s_c(t)) \iff c = c(t,s) = (s-1)e^{-\mu_{\mathcal{V}}t} + 1 - \int_0^t e^{-\mu_{\mathcal{V}}u}H(u)\mathrm{d}u,$$

and we deduce that (the initial condition is $g(0,s) = \phi_{\mathcal{U}}(s)$):

$$g(t,s) = g(0,c(t,s)) = \phi_{\mathcal{U}}(c(t,s)) = \phi_{\mathcal{U}}\left((s-1)e^{-\mu_{\mathcal{V}}t} + 1 - \int_0^t e^{-\mu_{\mathcal{V}}u}H(u)\mathrm{d}u\right). \tag{11}$$

**Solution of** (9). Let $s_\gamma(t) = \gamma e^{\mu_{\mathcal{V}}t} + 1$. Then, $s_\gamma'(t) = \mu_{\mathcal{V}}s(t) - \mu_{\mathcal{V}}$ and we deduce that, along the curves $(t,s_\gamma(t))$, $o(t,s)$ satisfies the following ODE:

$$\frac{\mathrm{d}}{\mathrm{d}t}o(t,s_\gamma(t)) = \frac{1 - q(t)^d}{1 - q(t)}\partial_s g(t,s_\gamma(t)).$$

Since

$$(t,s) = (t,s_\gamma(t)) \iff \gamma = \gamma(t,s) = (s-1)e^{-\mu_{\mathcal{V}}t},$$

we deduce that:

$$o(t, s) = \int_0^t H(u) \partial_s g(u, (s-1)e^{-\mu_{\mathcal{V}}(t-u)} + 1) \mathrm{d}u. \tag{12}$$

We now define the function $F(\cdot)$ as

$$F(t) := \int_0^t \mathrm{e}^{-\mu_{\mathcal{V}} u} H(u) \mathrm{d}u. \tag{13}$$

Using Equations (11) and (12), one can easily deduce that

$$\partial_s g(t, 1) = \mathrm{e}^{-\mu_{\mathcal{V}} t} \phi'_{\mathcal{U}}(1 - F(t))$$

and

$$\begin{aligned}
\partial_s o(t, 1) &= \int_0^t H(u) \mathrm{e}^{-\mu_{\mathcal{V}} u} \phi''_U(1 - F(u)) \mathrm{d}u \\
&= \phi'_U(1) - \phi'_U(1 - F(t)) = (\mu_{\mathcal{U}} - \phi'_{\mathcal{U}}(1 - F(t))) \mathrm{e}^{-\mu_{\mathcal{V}} t}.
\end{aligned}$$

In particular,

$$\partial_s g(t, 1) + \partial_s o(t, 1) = \partial_s f(\theta(t), 1) + \partial_s m(\theta(t), 1) = \mu_{\mathcal{U}} \mathrm{e}^{-\mu_{\mathcal{V}} t}.$$

Therefore,

$$H(t) = \frac{1 - \phi_{\mathcal{V}} \left( \frac{\partial_s o(t,1)}{\partial_s g(t,1) + \partial_s o(t,1)} \right)}{1 - \frac{\partial_s o(t,1)}{\partial_s g(t,1) + \partial_s o(t,1)}} = \mu_{\mathcal{U}} \frac{1 - \phi_{\mathcal{V}} \left( 1 - \frac{1}{\mu_{\mathcal{U}} \phi'_{\mathcal{U}}(1 - F(t))} \right)}{1 - \phi_{\mathcal{U}}(1 - F(t))},$$

which yields the following ordinary differential equation on $F$:

$$\frac{\frac{1}{\mu_{\mathcal{U}}} \phi'_{\mathcal{U}}(1 - F(t))}{1 - \phi_{\mathcal{V}} \left( 1 - \frac{1}{\mu_{\mathcal{U}}} \phi'_U(1 - F(t)) \right)} F'(t) = \mathrm{e}^{-\mu_{\mathcal{V}} t}. \tag{14}$$

## E  Proof of Theorem 2

We recall the notations introduced. For all $k \in \{0, \ldots, T\}$, all $c \in \{0, \ldots, C\}$ and all $i \geq 0$, we define:

- $F_i^{(c)}(k)$ the number of vertices of $\mathcal{U}$ that still have capacity $c$ at the end of step $k$ and whose remaining degree is $i$. Those vertices are referred to as *free* (with remaining degree $i$ and capacity $c$ at the end of step $k$).

- $M_i(k)$ the number of vertices of $\mathcal{U}$ that have capacity $c = 0$ at the end of step $k$ and whose remaining degree is $i$. Those vertices are referred to as *marked* (with remaining degree $i$ at the end of step $k$).

We also define as before the number of remaining half-edges to respectively free and marked vertices as

$$\widehat{F}(k) = \sum_{c=1}^C \sum_{i \geq 0} i F_i^{(c)}(k), \quad \text{and} \quad \widehat{M}(k) = \sum_{i \geq 0} i M_i(k).$$

The normalized performance of GREEDY is the ratio between the matched vertices in $\mathcal{V}$ and its maximal number, equal to $CN$:

$$A = \frac{\sum_{i \geq 0} \left( CM_i(T) + \sum_{c=1}^C (C - c) F_i^{(c)}(T) \right)}{CN}$$

As in the proof of Theorem 1:

1. we will place ourselves on the event where all vertices have bounded degrees, smaller than $N^\beta$ for some small $\beta > 0$

2. we will stop the analysis at $N^\gamma$ steps of the horizon $T$ so that $\widehat{F}(k) + \widehat{M}(k) > N^\gamma$ with arbitrarily high probability

3. we will distinguish the cases where $\widehat{M}(k) > 2N^\theta$ (with $\theta = \beta + \gamma/2$)

As a consequence, the errors are going to be of the same order of magnitude with the same order of probability (up to a multiplicative factor $C$) (hence those computations are skipped and replace by $\mathcal{O}(\cdot)$ notations). The interesting new component in this proof is the new system of differential equations and their solutions.

## E.1 The Differential equations

Using the same notations than in the proof of Theorem 1, we get that for all $0 \le k \le T$, $i \ge 0$ and $c \le C$,

$$
\mathbf{E}\big[F_i^{(c)}(k+1) - F_i^{(c)}(k) \,\big|\, \mathcal{F}_k\big]
$$

$$
= \mathbf{E}_{d_k \sim \pi_{\mathcal{V}}} \left[ \sum_{x=1}^{d_k} -q^{x-1} \frac{iF_i^{(c)}}{\widehat{F}+\widehat{M}} + p\sum_{x=1}^{d_k} q^{x-1}(d_k - x)\left( \frac{(i+1)F_{i+1}^{(c)} - iF_i^{(c)}}{\widehat{F}+\widehat{M}} \right) \right]
$$

$$
+ \mathbf{E}_{d_k \sim \pi_{\mathcal{V}}} \left[ \sum_{x=1}^{d_k} q^{x-1} \frac{(i+1)F_{i+1}^{(c+1)}}{\widehat{F}+\widehat{M}} \right] + \mathcal{O}(N^{\theta-\gamma})
$$

$$
= \frac{\mu_{\mathcal{V}}\left(-iF_i^{(c)} + (i+1)F_{i+1}^{(c)}\right) - (i+1)h(q)F_{i+1}^{(c)} + (i+1)h(q)F_{i+1}^{(c+1)}}{\widehat{F}+\widehat{M}} + \mathcal{O}(N^{\theta-\gamma})
$$

where the function $h$ is still defined as $h(q) = \frac{1-\phi_{\mathcal{V}}(q)}{1-q}$. Similarly, we can compute the expected increment in $M_i$ as

$$
\mathbf{E}\big[M_i(k+1) - M_i(k) \,\big|\, \mathcal{F}_k\big]
$$

$$
= \mathbf{E}_{d_k \sim \pi_{\mathcal{V}}} \left[ \left( \frac{d_k(1-p)^{d_k}}{\widehat{M}} + \sum_{x=1}^{d_k} pq^{x-1}\left( \frac{(x-1)}{\widehat{M}} + \frac{d_k - x}{\widehat{M}+\widehat{F}} \right) \right)\left((i+1)M_{i+1} - iM_i\right) \right]
$$

$$
+ \mathbf{E}_{d_k \sim \pi_{\mathcal{V}}} \left[ \sum_{x=1}^{d_k} pq^{x-1} \frac{(i+1)F_{i+1}^{(1)}}{\widehat{F}} \right] + \mathcal{O}(N^{\theta-\gamma})
$$

$$
= \frac{\mu_{\mathcal{V}}((i+1)M_{i+1} - iM_i) + (i+1)h(q)F_{i+1}^{(1)}}{\widehat{F}+\widehat{M}} + \mathcal{O}(N^{\theta-\gamma})
$$

From this, we get the following system of differential equations:

$$
\partial_t f^{(c)}(t,s) = \frac{1}{\mu_{\mathcal{U}} - t\mu_{\mathcal{V}}} \left[ \left(-\mu_{\mathcal{V}}s + \mu_{\mathcal{V}} - h(q(t))\right)\partial_s f^{(c)}(t,s) + \frac{1}{\mu_{\mathcal{U}} - t\mu_{\mathcal{V}}} h(q(t))\partial_s f^{(c+1)}(t,s) \right], \tag{15}
$$

and

$$
\partial_t m(t,s) = \frac{1}{\mu_{\mathcal{U}} - t\mu_{\mathcal{V}}} \left[ \left(-\mu_{\mathcal{V}}s + \mu_{\mathcal{V}}\right)\partial_s m(t,s) + h(q(t))\partial_s f^{(1)}(t,s) \right] \tag{16}
$$

With those notations, the normalized performances of GREEDY rewrite then into:

$$
A = m(\frac{\mu_{\mathcal{U}}}{\mu_{\mathcal{V}}}, 1) + \sum_{c=1}^{C} (1 - \frac{c}{C}) f^{(c)}(\frac{\mu_{\mathcal{U}}}{\mu_{\mathcal{V}}}, 1)
$$

$$
= 1 - \sum_{c=1}^{C} \frac{c}{C} f^{(c)}(\frac{\mu_{\mathcal{U}}}{\mu_{\mathcal{V}}}, 1)
$$

## E.2 Solving the PDEs

As in the previous section, we start with a time change. Let

$$\theta(t) = \frac{\mu_{\mathcal{U}}}{\mu_{\mathcal{V}}}\left(1 - e^{-\mu_{\mathcal{V}}t}\right) \tag{17}$$

so that $\theta'(t) = \mu_{\mathcal{U}} - \theta(t)\mu_{\mathcal{V}}$. In order to simplify notations, we set:

$$H(t) := h\left(q(\theta(t))\right). \tag{18}$$

Then, the new functions

$$g^{(c)}(t,s) := f^{(c)}(\theta(t),s) \qquad \text{and} \qquad o(t,s) := m(\theta(t),s)$$

satisfy the following PDEs:

$$\partial_t g^{(c)}(t,s) = \left[-\mu_{\mathcal{V}}s + \mu_{\mathcal{V}} - H(t)\right]\partial_s g^{(c)}(t,s) + H(t)\partial_s g^{(c+1)}(t,s),$$

and

$$\partial_t o(t,s) = \left[-\mu_{\mathcal{V}}s + \mu_{\mathcal{V}}\right]\partial_s o(t,s) + H(t)\partial_s g^{(1)}(t,s). \tag{19}$$

We distinguish:

$$\partial_t g^{(C)}(t,s) = \left[-\mu_{\mathcal{V}}s + \mu_{\mathcal{V}} - H(t)\right]\partial_s g^{(C)}(t,s) \tag{20}$$

We define:

$$F(t) = \int_0^t e^{-\mu_{\mathcal{V}}u} H(u)\mathrm{d}u$$

**Solution of (20).** This equation is the same as the one satisfied by $g(t,s)$, with the same initial conditions. Thus, we can write:

$$g^{(C)}(t,s) = \phi_{\mathcal{U}}\left((s-1)e^{-\mu_{\mathcal{V}}t} + 1 - F(t)\right).$$

**Solution of (20).** Lets define the curves:

$$s_{t,s}(u) = \left[(s-1)e^{-\mu_{\mathcal{V}}t} - F(t) + F(u)\right]e^{\mu_{\mathcal{V}}u} + 1.$$

Along those curves, we have:

$$\frac{\mathrm{d}}{\mathrm{d}t}g^{(c)}(u, s_{t,s}(u)) = H(u)\partial_s g^{(c+1)}(u, s_{t,s}(u)).$$

So:

$$g^{(c)}(t,s) = \int_0^t H(u)\partial_s g^{(c+1)}(u, s_{t,s}(u))\mathrm{d}u$$

**Solution for $c = C - 1$.** We have:

$$g^{(c-1)}(t,s) = \int_0^t H(u)\partial_s g^{(C)}(u, s_{t,s}(u))\mathrm{d}u$$

$$= \int_0^t F'(u)\phi'_U((s-1)e^{-\mu_{\mathcal{V}}u} + 1 - F(u))\mathrm{d}u$$

$$= F(t)\phi'_U((s-1)e^{-\mu_{\mathcal{V}}t} + 1 - F(t))$$

**Solution for $c = C - k$, general formula.** We will prove by induction:

$$g^{(C-k)}(t,s) = \frac{1}{k!}(F(t))^k \phi^{(k)}((s-1)e^{-\mu_{\mathcal{V}}t} + 1 - F(t))$$

If it is true for rank $k$, we have:

$$\partial_s g^{(C-k)}(u, s_{t,s}(u)) = \frac{e^{-\mu_{\mathcal{V}}u}}{k!}(F(u))^k \phi^{(k+1)}\left((s-1)e^{-\mu_{\mathcal{V}}t} + 1 - F(t)\right)$$

Which gives:

$$g^{(C-(k+1))}(t,s) = \frac{1}{k!}\left(\int_0^t F'(u)(F(u))^k \mathrm{d}u\right)\phi^{(k+1)}\left((s-1)e^{-\mu_{\mathcal{V}}t} + 1 - F(t)\right)$$

$$= \frac{1}{(k+1)!}(F(t))^{k+1}\phi^{(k+1)}((s-1)e^{-\mu_{\mathcal{V}}t} + 1 - F(t))$$

**Solution of** (19). Let's define the curves:

$$\gamma_{s,t}(u) = 1 + (s-1)e^{-\mu\nu(t-u)}$$

Along those curves:

$$\frac{\mathrm{d}}{\mathrm{d}u}o(u, \gamma_{t,s}(u)) = H(u)\partial_s g^{(1)}(u, \gamma_{t,s}(u))$$

So:

$$o(t,s) = \int_0^t F'(u)\frac{(F(u))^{(C-1)}}{(C-1)!}\phi^{(C)}\left((s-1)e^{-\mu\nu t} + 1 - F(u)\right)\mathrm{d}u$$

**Formula for GREEDY performances.** Recall that the normalized performances of GREEDY are

$$A = 1 - \sum_{c=1}^C \frac{c}{C}g^{(c)}(+\infty, 1)$$

$$= 1 - \sum_{k=0}^{C-1}\frac{1 - \frac{k}{C}}{k!}(F(+\infty))^k\phi^{(k)}(1 - F(+\infty))$$

## E.3  ODE for F

We have as before:

$$F'(t) = H(t)e^{-\mu\nu t}, \ H(t) = \frac{1 - \phi_{\mathcal{V}}(Q(t))}{1 - Q(t)}$$

And we also have:

$$Q(t) = \frac{\partial_s o(t,1)}{\partial_s o(t,1) + \sum_{c=1}^C \partial_s g^{(c)}(t,1)}$$

According to the previous section :

$$\partial_s o(t,1) = \left(\int_0^t F'(u)\frac{(F(u))^{(C-1)}}{(C-1)!}\phi_{\mathcal{U}}^{(C+1)}(1 - F(u))\mathrm{d}u\right)e^{-\mu\nu t}$$

$$= \left(\int_0^{F(t)}\frac{x^{(C-1)}}{(C-1)!}\phi_{\mathcal{U}}^{(C+1)}(1-x)\mathrm{d}x\right)e^{-\mu\nu t}$$

$$= \left[\phi_U'(1) - \phi_U'(1 - F(t)) - \sum_{k=1}^{C-1}\frac{F(t)^K}{k!}\phi_{\mathcal{U}}^{(k+1)}(1 - F(t))\right]e^{-\mu\nu t}$$

Which gives:

$$Q(t) = 1 - \frac{1}{\mu_{\mathcal{U}}}\left(\phi_U'(1 - F(t)) + \sum_{k=1}^{C-1}\frac{F(t)^k}{k!}\phi_{\mathcal{U}}^{(k+1)}(1 - F(t))\right)$$

We define:

$$\Gamma_U(F(t)) = \frac{1}{\mu_{\mathcal{U}}}\left(\phi_U'(1 - F(t)) + \sum_{k=1}^{C-1}\frac{F(t)^k}{k!}\phi_{\mathcal{U}}^{(k+1)}(1 - F(t))\right)$$

This yields the following differential equation for $F$:

$$\frac{\Gamma_U(F(t))}{1 - \phi_{\mathcal{V}}(1 - \Gamma_U(F(t)))}F'(t) = \mathrm{e}^{-\mu\nu t}.$$

Theorem 2 then follows from the same arguments in the proof of Theorem 1 (except that errors are $C$ times bigger as there are $C$ more equations to handle).

# F    Proof of Theorem 3

As mentioned in the main text, the only difference with Theorem 2 is that $C$ could be of the order of $N^\beta$ (but not bigger on the event where all degrees are smaller than $N^\beta$). As a consequence, one must take $\beta$ even smaller than $1/20$ to have sublinear errors terms (choosing $\beta = 1/40$ is admissible for instance) with exponentially high probability.

**Solution of (20).**   This equation is the same as the one satisfied by $g(t,s)$, the new initial condition is $g^{(C)}(t,s) = p_C \phi_{\mathcal{U}}(s)$. Thus, we can write:

$$g^{(C)}(t,s) = p_C \phi_{\mathcal{U}}\left((s-1)e^{-\mu\nu t} + 1 - F(t)\right).$$

**Solution for $c = C - 1$.**   We have:

$$g^{(c-1)}(t,s) = \int_0^t H(u)\partial_s g^{(C)}(u, s_{t,s}(u))\mathrm{d}u + g^{(c-1)}(0, s_{t,s}(0))$$

$$= \int_0^t F'(u)\phi_U'((s-1)e^{-\mu\nu t} + 1 - F(t))\mathrm{d}u + p_{(C-1)}\phi_{\mathcal{U}}((s-1)e^{-\mu\nu t} - F(t) + 1)$$

$$= p_C F(t)\phi_U'((s-1)e^{-\mu\nu t} + 1 - F(t)) + p_{(C-1)}\phi_{\mathcal{U}}((s-1)e^{-\mu\nu t} + 1 - F(t))$$

**Solution for $c = C - 2$.**

$$\partial_s g^{(C-2)}(u, s_{t,s}(u)) = p_C e^{-\mu\nu u}F(u)\phi_{\mathcal{U}}''\left((s-1)e^{-\mu\nu t} + 1 - F(t)\right) + p_{(C-1)}e^{-\mu\nu u}\phi_{\mathcal{U}}'(s_{t,s}(u))$$

Let's define:

$$c(t,s) = (s-1)e^{-\mu\nu t} + 1 - F(t)$$

Which gives:

$$g^{(C-2)}(t,s) = p_{(C-1)}\left(\int_0^t F'(u)F(u)\mathrm{d}u\right)\phi''(c(t,s))$$

$$+ p_{(C-1)}\left(\int_0^t F'(u)e^{\mu\nu u}\mathrm{d}u\right)\phi'(c(t,s))\,\mathrm{d}u + p_{(C-2)}\phi_{\mathcal{U}}(s_{t,s}(0))$$

$$= \frac{p_C}{2}(F(t))^2\phi''((s-1)e^{-\mu\nu t}$$

$$+ 1 - F(t)) + p_{(C-1)}F(t)\phi'((s-1)e^{-\mu\nu t} + p_{(C-2)}\phi_{\mathcal{U}}(c(t,s))$$

**Solution for $c = C - k$, general formula.**   We will prove by induction:

$$g^{(C-k)}(t,s) = \sum_{l=0}^{k} p_{C-l}\frac{1}{(k-l)!}(F(t))^{k-l}\phi^{(k-l)}(c(t,s))$$

If it is true for rank $k$, we have:

$$\partial_s g^{(C-k)}(u, s_{t,s}(u)) = \sum_{l=0}^{k} p_{C-l}\frac{1}{(k-l)!}(F(t))^{k-l}e^{-\mu\nu u}\phi^{(k+1-l)}(c(t,s))$$

Which gives:

$$g^{(C-(k+1))}(t,s) = p_{(C-(k+1))}\phi_{\mathcal{U}}(c(t,s)) + \sum_{l=0}^{k} p_{(C-l)}\frac{1}{(k-l)!}\left(\int_0^t F'(u)(F(u))^{(k-l)}\mathrm{d}u\right)\phi^{(k+1-l)}(c(t,s))$$

$$= \sum_{l=0}^{k+1} p_{(C-l)}\frac{1}{(k+1-l)!}(F(t))^{k+1-l}\phi^{(k+1-l)}(c(t,s))$$

**Solution of** (19).

$$o(t, s) = \sum_{c=1}^{C} p_c \int_0^t F'(u) \frac{(F(u))^{(c-1)}}{(c-1)!} \phi^{(c)} \left((s-1)e^{-\mu_\mathcal{V} t} + 1 - F(u)\right) \mathrm{d}u$$

$$g^{(c)}(t, s) = \sum_{k=0}^{C-c} p_{c+k} \frac{1}{k!} (F(t))^k \phi^k(c(t, s))$$

**Quantity of interest.**

$$A = \frac{\mu_\mathcal{V}}{\mu_\mathcal{U}} \sum_{c=1}^{C} c(p_c - g^{(c)}(+\infty, 1))$$

$$= \frac{\mu_\mathcal{V}}{\mu_\mathcal{U}} \left( \sum_{c=1}^{C} c p_c - \sum_{k=0}^{C-1} \left( \frac{1}{k!} (F(+\infty))^k \phi^{(k)} \left(1 - F(+\infty)\right) \sum_{c=1}^{C} c p_{c+k} \right) \right)$$

**ODE for the function F.**

$$\partial_s o(t, 1) = \left( \sum_{c=1}^{C} p_c \int_0^t F'(u) \frac{(F(u))^{(c-1)}}{(c-1)!} \phi_\mathcal{U}^{(c+1)} \left(1 - F(u)\right) \mathrm{d}u \right) e^{-\mu_\mathcal{V} t}$$

$$= \left[ \phi_U'(1) - \phi_U'(1 - F(t)) + \sum_{k=1}^{C-1} \left( \frac{F(t)^k}{k!} \phi_\mathcal{U}^{(k+1)}(1 - F(t)) \sum_{c=k+1}^{C} p_c \right) \right] e^{-\mu_\mathcal{V} t}$$

Which yields:

$$Q(t) = 1 - \frac{1}{\mu_\mathcal{U}} \left( \phi_U'(1 - F(t)) + \sum_{k=1}^{C-1} \left( \frac{F(t)^k}{k!} \phi_\mathcal{U}^{(k+1)}(1 - F(t)) \sum_{c=k+1}^{C} p_c \right) \right)$$

We define:

$$\Gamma_U(F(t)) = \frac{1}{\mu_\mathcal{U}} \left( \phi_U'(1 - F(t)) + \sum_{k=1}^{C-1} \left( \frac{F(t)^k}{k!} \phi_\mathcal{U}^{(k+1)}(1 - F(t)) \sum_{c=k+1}^{C} p_c \right) \right)$$

This yields the following differential equation for $F$:

$$\frac{\Gamma_U(F(t))}{1 - \phi_\mathcal{V} \left(1 - \Gamma_U(F(t))\right)} F'(t) = \mathrm{e}^{-\mu_\mathcal{V} t}.$$

# G    Proof of Proposition 1

**Lemma 4.** *On the $2$-regular graph, the law of the matches generated by the algorithm Ranking equals the law of the matches generated by a biased Greedy algorithm, that chooses a free vertex of degree $2$ over one of degree $1$ with probability at least $2/3$. This is biased as the classical Greedy algorithm chooses it with probability $1/2$.*

*Proof*: Two vertices of the same degree are interchangeable, they are both equally likely to have the smallest rank. Thus Ranking and Greedy behave the same on arriving vertices whose potential neighbors all have the same degree. Let $r(v)$ be the rank of vertex $v$ and $deg(v)$ its residual number of unpaired half-edges.

Let $\mathcal{A}_t(u, 2)$ be the following event:

- $u$ had one of its half-edges paired to the incoming vertex $v_t$ at iteration $t$,

- $u$ was not matched and $v_t$ was instead matched to a vertex of residual degree 2.

Let $\mathcal{A}_t(u, 1)$ be the similar event with $v_t$ instead matched to a vertex of residual degree 1.

$$\mathbf{P}\left(r(u) \geq k | \deg(u) = 1 \text{ and } u \text{ free at } t\right) = \sum_{t' < t} \mathbf{P}\left(r(u) \geq k | \mathcal{A}_{t'}(u, 2)\right) \mathbf{P}(\mathcal{A}_{t'}(u, 2) | \deg(u) = 1 \text{ and } u \text{ free at } t)$$

$$+ \sum_{t' < t} \mathbf{P}\left(r(u) \geq k | \mathcal{A}_{t'}(u, 1)\right) \mathbf{P}(\mathcal{A}_{t'}(u, 1) | \deg(u) = 1 \text{ and } u \text{ free at } t)$$

Now, assume

$$\forall t' < t, \forall a \in [N], \mathbf{P}(r(a) \geq k | \deg(a) = 1 \text{ and } a \text{ free at } t) \geq \mathbf{P}(r(a) \geq k | \deg(a) = 2). \quad (21)$$

Hypothesis 21 implies:

$$\forall t' < t, \mathbf{P}(r(u) \geq k | \mathcal{A}_{t'}(u, 1)) \geq \mathbf{P}(r(u) \geq k | \mathcal{A}_{t'}(u, 2)),$$

thus, the following inequality holds:

$$\mathbf{P}(r(u) \geq k | \deg(u) = 1 \text{ and } u \text{ free at } t) \geq \mathbf{P}(r(u) \geq k | \cup_{t' < t} \mathcal{A}_{t'}(u, 2)).$$

Let $a$ and $b$ be two different numbers randomly chosen in $[n]$. Two vertices with two remaining half-edges were not affected by the run before, and thus could have any rank. It therefore holds:

$$\mathbf{P}\left(r(u) \geq k | \deg(u) = 1 \text{ and } u \text{ free at } t\right) \geq \mathbf{P}\left(\max(a, b) \geq k\right)$$

$$= 1 - \frac{\binom{k-1}{2}}{\binom{n}{2}}$$

$$= 1 - \frac{(k-1)(k-2)}{n(n-1)}.$$

This inequality implies $\mathbf{P}\left(r(u) \geq k | \deg(u) = 1 \text{ and } u \text{ free at } t\right) \geq \mathbf{P}\left(r(u) \geq k | \deg(u) = 2\right)$, thus 21 is true by induction.

Therefore, vertices with only one remaining half-edges are likely to have a higher rank than those with two remaining half-edges:

$$\mathbf{P}(r(b) < r(a) | \deg(b) = 2, \deg(a) = 1) = \sum_{k=1}^{n} \mathbf{P}(\deg = k - 1, \deg(a) \geq k | \deg(b) = 2, \deg(a) = 1)$$

$$\geq 1 - \sum_{k=1}^{n} \frac{(k-1)(k-2)}{n^2(n-1)}$$

$$\geq \frac{2}{3} + O\left(\frac{1}{n}\right).$$

$\square$

Let $M_1^G(t)$ and $M_1^R(t)$ be the number of marked vertices of degree 1 by GREEDY and RANKING algorithms respectively. Note that the number of vertices of degree 2 is the same for both algorithms, $F_2^G(t) = F_2^R(t)$. Also, the following always holds

$$F_1^G(t) = 2N - 2t - 2F_2^G(t) - M_1^G(t).$$

This shows that in 2-regular graphs, the number of half-edges in each ensemble is a deterministic quantity of $M_1^R(t + 1)$ and $M_1^G(t + 1)$.

Suppose it holds at time $t$ that $M_1^G(t) = M_1^R(t) = M_1(t)$ (event $\mathcal{A}$), then

$$\mathbb{E}[M_1^R(t+1)|\mathcal{A}] - \mathbb{E}[M_1^G(t+1)|\mathcal{A}] = \mathbb{E}[\mathbb{1}_{\{\text{RANKING marks a vertex in } F_2^R(t)\}}|\mathcal{A}] - \mathbb{E}[\mathbb{1}_{\{\text{GREEDY marks a vertex in } F_2^R(t)\}}|\mathcal{A}]$$

$$= \frac{1}{6} \cdot \frac{F_1(t)F_2(t)}{2(N-t)} > 0$$

The first equality holds since the probability of pairing an half-edge in $M_1^R(t+1)$ and $M_1^G(t+1)$ only depends on the number of half edges in each ensembles, not on the algorithm.

Therefore, by application of Gronwald's lemma, RANKING generates strictly more marked vertices of degree $1$. As the probability that an incoming vertice is matched only to non-available vertices increases with $M_1$, RANKING performs strictly worse than GREEDY on 2-regular graphs.