# OpenReview forum: "Online Matching in Sparse Random Graphs: Non-Asymptotic Performances of Greedy Algorithm"
_NeurIPS.cc/2021/Conference — NeurIPS 2021 Poster_

### Official Review · Reviewer_mSP9 · 2021-07-16

**Rating:** 6
**Confidence:** 2

**Summary:**

This paper analyzes the performance of the GREEDY algorithm for online matching on random bipartite graphs generated by the configuration model with given degree distributions.
The analysis is based the differential equation method, a stochastic approximation using an ODE to represent the time evolution of matching size under the GREEDY algorithm with overwhelming probability.
The paper reports the following main results:
- A proof that the matching size of the GREEDY algorithm is the solution to an ODE.
- Competitive ratios of the GREEDY algorithm in different graph models (d-regular graphs, Erdos-Reny graphs).
- A proof that GREEDY beats RAKNING on 2-regular graph, with experimental evidence suggesting that the same holds for higher degrees.

**Limitations And Societal Impact:**

This paper only studies theoretical performance and does not mention societal impacts.

**Main Review:**

- Very readable, great writing, though typos are noticeable.
- Main idea is to consider the random process of graph generation in the configuration model together with the time evolution of GREEDY matching, which is interesting.
- The main result generalizes or complements some known results on special graph families.
- The paper is easy to follow.

Some nitpicks:
- Typo: line 19, Page 1, second paragraph of Introduction, Internal advertising should be Internet advertising.
- Typo: Algorithm ??, Page 4, line 129.
- Great introduction on online matching and its variants (weighted, configuration model).

**Time Spent Reviewing:**

3

---

### Official Review · Reviewer_v6Xy · 2021-07-17

**Rating:** 5
**Confidence:** 3

**Summary:**

The authors investigate bipartite matching in an online setting were the graph edges are generated via the configuration model (i.e. from a chosen degree distribution). They bound the estimation error, measured via the expected ratio between the online matching obtained by Greedy and the hindsight-optimal matching. This allows them to compare the performance guarantee for Greedy versus the state-of-the-art Ranking algorithm. The authors prove that surprisingly, there exist problem instances (including simple and natural ones) where Greedy outperforms Ranking.

**Limitations And Societal Impact:**

The limitations are addressed. Societal impact is not applicable.

**Main Review:**

REVIEW
The main result, which is a bound on the estimation error of Greedy for configuration graphs, is of theoretical interest. It is also interesting that this result permits a convenient comparison of Greedy's performance on different configuration graphs. Finally, I agree with the authors that it is interesting that Greedy outperforms Ranking on simple/natural graphs, such as 2-regular graphs. Obtaining these results is nontrivial. I see this not as a breakthrough theoretical result, but rather as an interesting step forward in a relevant area that requires nontrivial analysis.

In terms of style, the paper is generally well-written. However, typical NeurIPS papers make better use of subheadings throughout and offer brief exposition/high-level intuitions around each section. This paper is written more like a math journal paper. More structure and more high-level exposition would improve the writeup.

The experimental results figures are in general very difficult to parse and have some issues: They are missing labels on the axes; linetypes (dashes/dots) are not used to distinguish the lines; points are not used to show the actual points of comparison; I infer that (for example) the 4 panels in Appendix Fig. 5 and 6 are used to plot d=2,4,10,20 (this should be stated), but there are no titles on the figures, so is the top right plot d=4 or the bottom left plot d=4?

Also, in the figures shown in the paper's text (Fig. 1), we are shown results for d=[2,3,4,6,10], and we are told results are nearly instantaneous. Given that computation is not a limiting factor, why not show us more results (in a different type of plot)? In general, for a NeurIPS paper I would expect a greater effort to articulate and display experimental results---improving this will improve the paper. Also, one of the last sentences of the paper mentions that you expect Greedy will outperform Ranking in various other scenarios---this is a tantalizing opportunity for a broader experimental approach (either for this or another paper).

I will consider revising my review based on the authors' response.

Minor suggestions for improving the writing, figures, etc.:

- References are all lacking numbers in the References section. This makes it very difficult to check your references.

-Line 69 states "The main is that " --> maybe there is a missing word such as 'main limitation'?

-Convention suggests that your introduction should have subsections. It is a well-written introduction, but we only learn about your main result in the penultimate paragraph. You might consider highlighting it under a "Main contribution" substitle or bold paragraph heading.

- The Algorithm \ref in line 129 is undefined.

- Multiple instances of the word GREEDY have a missing space after. If this is a macro, try a \ (slash) between GREEDY and any non-punctuation letters/words.

- Missing 's' in 'situations' on line 308

- Line 212: folline -> offline

**Time Spent Reviewing:**

3

---

> ### Author Response · Authors · 2021-08-09
> **Additional experiments on Greedy vs. Ranking and structure improvements**
>
> We thank you for the time spent on the paper and your constructive feedback.
> We greatly appreciate your comments on experimental results. We will include your layout comments on the figures as we agree that this should improve clarity.
> As for the additional experimental results on the relative performance of Greedy vs. Ranking, we have conducted a broader exploration and will happily include the following plots:
> - Greedy and Ranking have the same performance on Erdos-Renyi graphs (which is explained in theory by the memory-less property of those graphs).
> - Greedy outperforms Ranking in d-regular graphs. Clearer plots supporting this claim will be moved from the appendix to the main body of the text.
> - Ranking outperforms Greedy graphs with a group of low degree vertices and a group of high degree vertices (for instance $\pi_u(3)=0.5$, $\pi_u(9)=0.5$, $\pi_u=\pi_v$).
> A precise criteria for the relative shift in performance of Greedy and Ranking would be an interesting result left for future work.
>
> We also thank you for your pointers on the structure of the paper. In the camera ready we will gladly take them into account, restructuring the introduction, and will make use of the extra page to include additional high-level intuitions.

---

> > ### Author Response · Authors · 2021-09-02
> > **Remaining questions ?**
> >
> > Dear Reviewer v6Xy, you had some questions and concerns we tried to answer.  In your review, you mentioned that you would consider revising your score based on our response. We did not hear back from you, we were wondering if that means you still have questions (feel free to ask !) or if you are now convinced; in the latter case, we hope it’s enough for you to reassess your score as you indicated.

---

### Official Review · Reviewer_8M7X · 2021-07-19

**Rating:** 5
**Confidence:** 3

**Summary:**

The paper studies the online bipartite matching problem with capacities in random graphs that are generated by a stochastic process called the configuration model. The configuration model generates random graphs whose degree distributions are sub-Gaussian. The main contribution of the paper is an analysis of the competitive ratio achieved by the well-studied Greedy algorithm for online bipartite matching. The main result relates the competitive ratio to the solution of a certain ODE. The ODE can either be solved in closed form for some special cases of interest or it can be solved numerically to obtain estimates of the competitive ratio. As a corollary of the main result, the paper derives competitive ratios for several special cases, including random d-regular graphs and Erdos-Renyi graphs.


**Limitations And Societal Impact:**

Yes

**Main Review:**

Significance: The paper addresses a fundamental problem in online algorithms, which is well studied and has applications in online advertising. The Greedy algorithm is also a well-studied algorithm in both the adversarial setting as well as stochastic settings. The current paper contributes to this line of work by analyzing Greedy in a more general random graph models, and relating its competitive ratio by relating it to an ODE that can be solved either numerically or in closed form in some settings. The questions addressed are mathematically interesting.

On the negative side, the paper does not discuss how the mathematical results presented can benefit the machine learning community. The paper does not connect the random graph model considered to real-world instances. Overall, the paper seems to be a better fit for a TCS or Mathematics venue.

Novelty/originality: The result appears to be novel and the approach could potentially lead to explicit competitive ratios for more general random instances.

Clarity: The content in the main body is reasonably clear. I have not checked the correctness of the proofs included in the appendix.

**Time Spent Reviewing:**

I did not track the hours

---

> ### Author Response · Authors · 2021-08-09
> **Benefits of our work to the machine learning community and connections with real-world instances**
>
> Let us first thank you for the time you spent reading the article and for your constructive feedback. As pointed out, we did not mention the benefits of our work to the machine learning community and its connections with real-world instances. We agree this could improve the exposition and will be happy to add the following comments:
> - The configuration model is a well-suited random graph model which mimics a number of properties of real-world complex networks, while being analytically tractable. For instance, choosing power-law distributions for the degrees allows to obtain the so-called scale-free property (often observed in practice, as highlighted for the web by Faloustos and al. in On power-law relationships of the internet topology ; Computer Communications Rev. ; 1999). The configuration model also displays the so called “small-world phenomenon” (observed for instance in the graph of Facebook by Backstrom and al. in Four degrees of separation ; In Proceedings of the 4th Annual ACM Web Science Conference ; 2012) as its diameter is of logarithmic order.
>
> - We believe online matching will become an important problem for the machine learning community in the future. Each year, the complexity of the underlying graphs increases and we are considering adding features to the model in future work (such as random variables on the edges, modeling the interest for a consumer for a given product), or connection modeled via some Kernel between vertices features (say, if users & products/campaigns are embedded in the same space). In this context, machine learning tools will certainly be needed to tackle the problem.
>
> There already exists an active community of researchers working on online matching in probability, statistics, computer science and economics. We believe there is a great opportunity for the machine learning community to make substantial contributions to the area. In this regard, our paper can also be thought of as a first step and introduction to the subject.

---

### Official Review · Reviewer_tZCg · 2021-07-19

**Rating:** 7
**Confidence:** 3

**Summary:**

The paper studies the performance of a greedy algorithm for online bipartite matching under the configuration model. In this model the vertex degrees of the input graph adhere to a fixed distribution. The authors estimate the competitive ratio of the greedy algorithm and show that in this model the greedy algorithm achieves better performance guarantees than related methods.

**Limitations And Societal Impact:**

No societal impact.

**Main Review:**

Originality/Significance: The authors present an interesting result in the area of online matching under a stochastic model. It has to be said that I am not an expert in this area, but it seems both the result and the proof technique are original and should be of interest to the community.

Quality/Clarity: The paper is mostly clear. At times the authors use terminology that could be improved, such as:
- "subGaussian" (line 113) -- What do you mean by that?
- "half-edges" -- requires definition
- "multiple edges" (line 158) -- Should be replaced by "parallel edges" (the standard terminology)
- The statement in Proposition 1 should be formalized.

I did not have time to check the proof of the main claim.

From a presentation point of view the paper should meet the bar of acceptance, given the authors improve on the issues pointed out above.

Typos:
- $N$ instead of $n$ (line 164)
- *reason (line 69)
- "worst" -> "worse" (line 214)


**Time Spent Reviewing:**

3

---

> ### Author Response · Authors · 2021-08-09
> **Terminology fixes and Prop.1 reformulation**
>
> Thank you for pointing out those terminology points, fixing them will certainly improve the clarity of the paper.
> We will rephrase Proposition 1 for the camera ready.

---

### Decision · Program_Chairs · 2021-09-27

**Decision:**

Accept (Poster)

**Comment:**

The reviewers generally thought this was an interesting paper on stochastic matchings. Novel ideas and analysis are presented.  The criticism is the fit for the community.